# DNA methylation as a mediator of *HLA-DRB1*15:01* and a protective variant in multiple sclerosis

Lara Kular et al.[#]

The human leukocyte antigen (HLA) haplotype *DRB1*15:01* is the major risk factor for multiple sclerosis (MS). Here, we find that *DRB1*15:01* is hypomethylated and predominantly expressed in monocytes among carriers of *DRB1*15:01*. A differentially methylated region (DMR) encompassing *HLA-DRB1* exon 2 is particularly affected and displays methylation-sensitive regulatory properties in vitro. Causal inference and Mendelian randomization provide evidence that *HLA* variants mediate risk for MS via changes in the *HLA-DRB1* DMR that modify *HLA-DRB1* expression. Meta-analysis of 14,259 cases and 171,347 controls confirms that these variants confer risk from *DRB1*15:01* and also identifies a protective variant (rs9267649, $p < 3.32 \times 10^{-8}$, odds ratio $= 0.86$) after conditioning for all MS-associated variants in the region. rs9267649 is associated with increased DNA methylation at the *HLA-DRB1* DMR and reduced expression of *HLA-DRB1*, suggesting a modulation of the *DRB1*15:01* effect. Our integrative approach provides insights into the molecular mechanisms of MS susceptibility and suggests putative therapeutic strategies targeting a methylation-mediated regulation of the major risk gene.

[#]A full list of authors and their affiliations appears at the end of the paper. Correspondence and requests for materials should be addressed to A.P.F. (email: afeinberg@jhu.edu) or to M.J. (email: Maja.Jagodic@ki.se)

Multiple sclerosis (MS), a leading cause of neurological disability in young adults, is a chronic inflammatory disease of the central nervous system (CNS) characterized by autoimmune destruction of myelin and subsequent loss of neurons. Although the exact cause of MS remains unknown, inheritance of the disease is consistent with one locus exerting a moderate effect and many loci with modest effects[1]. The first genetic risk factor was established more than 40 years ago in the human leukocyte antigen (HLA) locus[2], which encodes molecules involved in key immune functions. The HLA genes are among the most polymorphic genes and several alleles are often inherited together in extended haplotypes due to extremely high linkage disequilibrium (LD) in this part of the genome. The extended haplotype of the HLA class II region (DRB5*01:01-DRB1*15:01-DQA1*01:02-DQB1*06:02), which has been further refined to DRB5*01:01-DRB1*15:01[3–5], confers the strongest risk for developing MS[6]. HLA class II locus encodes molecules involved in presentation of peptide antigens to T cells by antigen presenting cells (APCs) and DRB1*15:01 confers a 3-fold increased risk of developing MS. With the advent of genome-wide association studies (GWAS) more than 100 additional non-HLA loci have been identified predisposing for MS with modest effects[4,7–10]. However, the MS risk loci identified to date explain only about half of the disease heritability[8] and little is known about the underlying causal variants and their molecular mechanisms.

Recent genetic and epigenetic fine-mapping efforts suggest that a vast majority of causal candidate variants for autoimmune diseases are non-coding and likely play a role in regulating gene expression[11]. Epigenetic mechanisms can regulate gene expression by modification of DNA in a manner that is heritable through cell divisions. The most studied epigenetic mechanism is the covalent addition of a methyl group to cytosines in the context of CpG dinucleotides, because of a known stable mechanism for the propagation of $^mCpG$ by DNA (Cytosine-5)-Methyltransferase 1. Alterations in DNA methylation have been reported in blood, $CD4^+$, and $CD8^+$ T cells as well as in pathology-free brain regions from MS patients[12–15]. Recently, genetic variants in the loci encoding epigenetic machinery genes have been associated with MS suggesting a role for epigenetic mechanisms in disease pathology[9,16]. However, while studies have investigated genetic and epigenetic mechanisms independently, little focus has been on how they may interact at a locus-specific level and jointly affect susceptibility to MS. Indeed, a growing body of evidence suggests that genetic and epigenetic modifications can interact biologically[17,18]. This paradigm has been instrumental in deciphering the contribution of DNA methylation to the genetic risk that predisposes to other complex diseases. Several studies suggest that DNA methylation in the HLA class II region could mediate genetic susceptibility to immune-mediated diseases such as rheumatoid arthritis (RA)[19], type 1 diabetes[20], and food allergy[21].

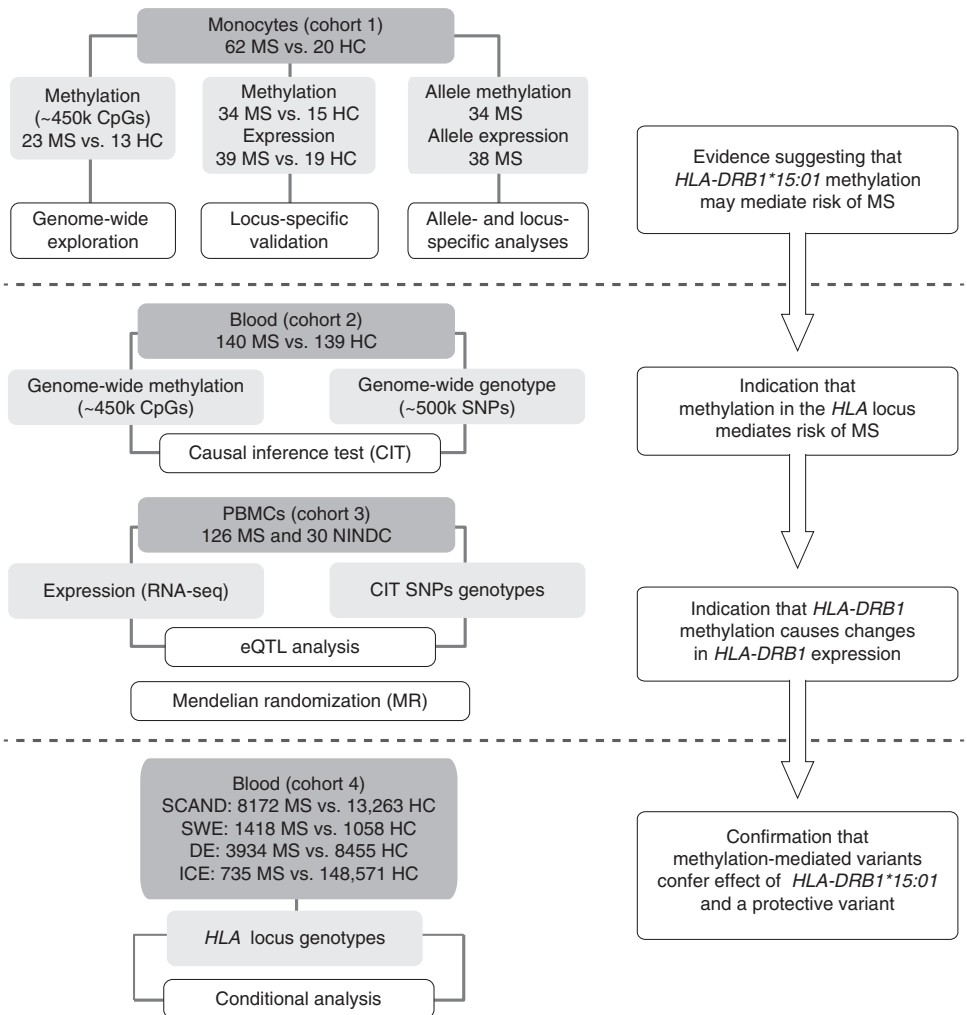

**Fig. 1** Study design and workflow diagram. MS: multiple sclerosis, HC: healthy controls, SNP: single nucleotide polymorphism, CIT: causal inference test, eQTL: expression quantitative trait loci, NINDC: non-inflammatory neurological disease controls, SCAND: Scandinavia, SWE: Sweden, DE: Germany, ICE: Iceland

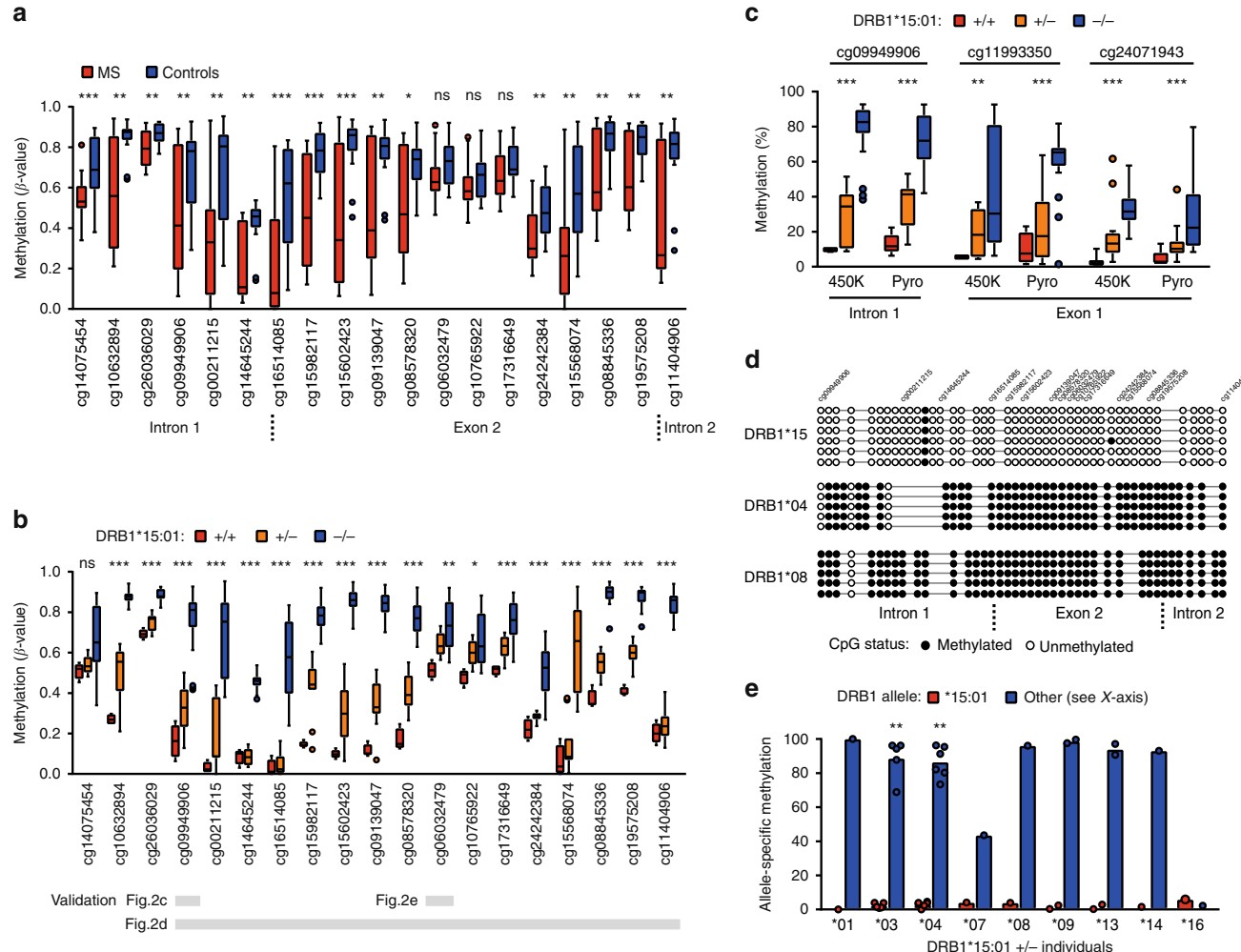

**Fig. 2** *DRB1*15:01*-associated DNA methylation in monocytes. **a** Differentially methylated CpGs using 450K arrays between multiple sclerosis (MS, n = 23, red) cases and healthy controls (HC, n = 13, blue) comprising two significant differentially methylated regions (Supplementary Data 2) in the *HLA-DRB1* gene. **b** Differential methylation according to *DRB1*15:01* haplotype: homozygous (+/+, n = 4, red), heterozygous (+/−, n = 11, orange), and non-carriers (−/−, n = 20, blue) in MS and HC together (p-values from the additive model). All probes and p-values from other models are shown in Supplementary Data 2. The horizontal gray bars indicate CpGs for which methylation has been validated using other methods (results depicted in **c–e**). **c** Replication using pyrosequencing of three CpGs according to *DRB1*15:01* haplotype: homozygous (+/+, n = 5, red), heterozygous (+/−, n = 17, orange), and non-carriers (−/−, n = 27, blue) (p-values were generated using Kruskal–Wallis test). **d** Results from Sanger sequencing of bisulfite PCR clones for exon 2 region of *HLA-DRB1*. Each line represents one read where black and white circles illustrate methylated and unmethylated CpGs, respectively (corresponding probes ID from the 450K array annotation are included). **e** Methylation of cg06032479 of each allele from individuals heterozygous for *DRB1*15:01* (n = 20, red and blue colors representing *DRB1*15:01* and the other allele, respectively) quantified by MSRE followed by allele-specific qPCR and tested using the Mann–Whitney test (for n > 3 individuals/group). Individuals' data are shown in Supplementary Fig. 1. **a–c** Data are presented as Tukey boxplots; *p < 0.05 **p < 0.01, ***p < 0.001

In this study, we investigate DNA methylation in MS patients in the context of genetic variation and gene expression with the aim to decipher biological consequences of inheritance of MS risk alleles and demonstrate that DNA methylation mediates risk of developing MS. Specifically, DNA methylation in the *HLA-DRB1* gene mediates the effect of the strongest MS risk variant *HLA-DRB1*15:01*, and of a protective *HLA* variant (rs9267649) which has not been previously reported, on *HLA-DRB1* expression and the risk of MS. Our results are summarized in Fig. 1.

## Results

### *DRB1*15:01* is hypomethylated and predominantly expressed.
We conducted DNA methylation analysis in monocytes sorted from MS patients and matched controls (n = 36, Fig. 1, cohort 1, Supplementary Data 1) using Illumina Infinium

HumanMethylation450 BeadChip arrays (450K arrays). Monocytes are important precursors of tissue APCs[22] and they have been implicated in MS through their ability to present myelin antigens, produce pro-inflammatory mediators, and phagocytose myelin[23–25]. We identified two differentially methylated regions (DMRs) that are associated with MS after adjustment for confounders (FWER < 0.05) (Supplementary Data 2). Both DMRs mapped to the *HLA-DRB1* gene and comprised 19 consecutive CpGs, encompassing exon 2, which were hypomethylated in MS patients (Fig. 2a). Since the *DRB1*15:01* allele of the *HLA-DRB1* gene confers the strongest risk for developing MS, we analyzed methylation differences between *DRB1*15:01* carriers and non-carriers. Homozygous *DRB1*15:01* carriers displayed significantly lower DNA methylation levels at *HLA-DRB1* compared to heterozygous carriers and non-carriers (Fig. 2b, Supplementary Data 2).

A potential bias in estimating DNA methylation with 450K arrays can arise from the impact of single nucleotide polymorphisms (SNPs). This is of particular importance in the *HLA* region due to its high density of polymorphic sites and the sequence similarity between proximal *HLA* genes. To rule out the possibility that low DNA methylation in *DRB1*15:01* carriers resulted from SNPs that abolish CpG sites or affect hybridization of the probes, we used three different approaches (Fig. 2c–e). We replicated 450K array findings using locus-specific

pyrosequencing of bisulfite converted (BS)-DNA for selected CpGs for which robust assays could be designed (Fig. 2c, n = 49). We further assessed DNA methylation by cloning and sequencing single-strand BS-DNA from a larger segment encompassing exon 2 of *HLA-DRB1*. Sequencing results from five homozygous *DRB1*15:01* carriers confirmed that the CpG sites are not disrupted by local SNPs and that these CpGs preferentially exist in an unmethylated state (Fig. 2d, only one carrier shown) in the *DRB1*15:01* carriers, compared to carriers of other haplotypes,

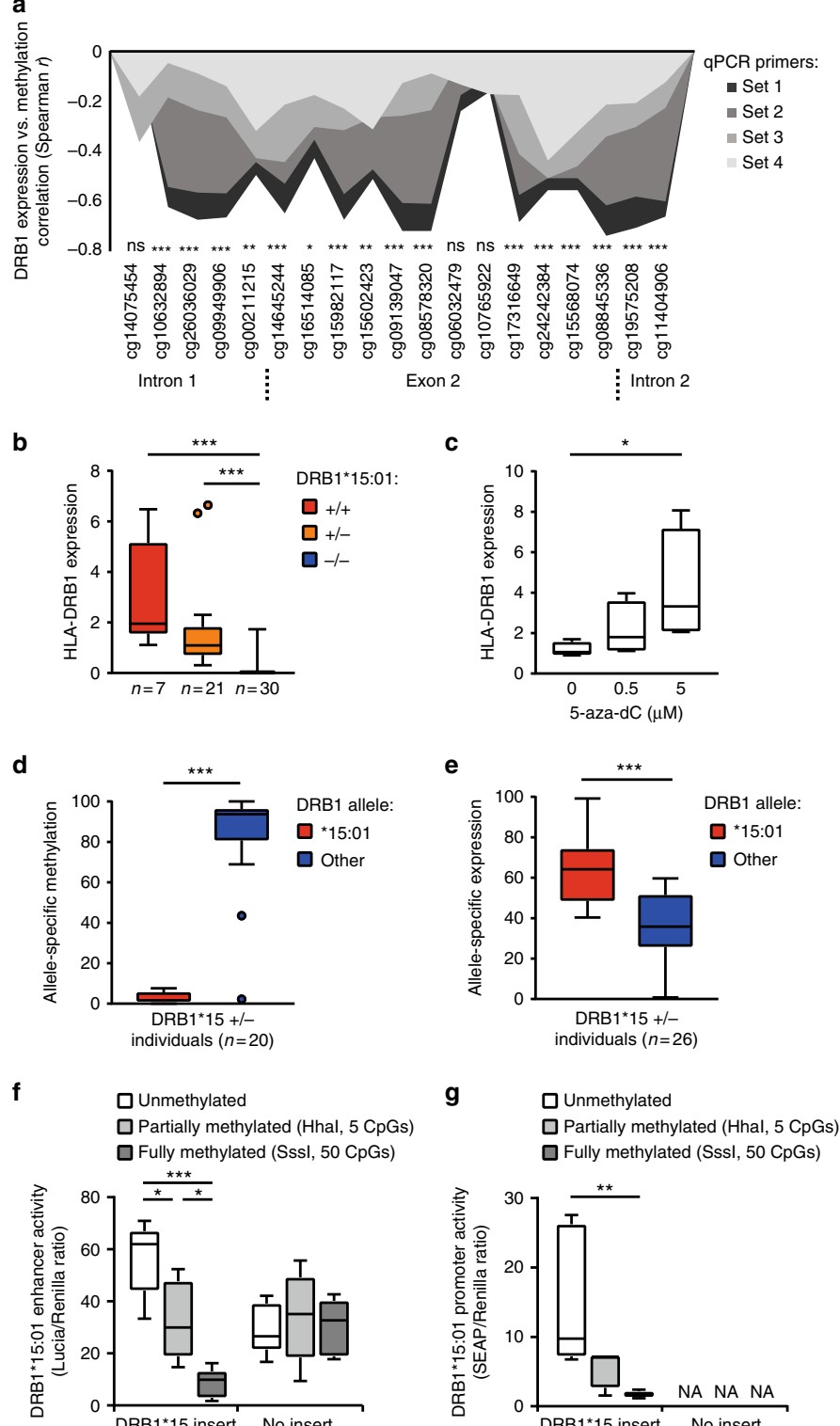

e.g., *DRB1*04* and *DRB1*08*. We also addressed allele-specific methylation at cg06032479 using methyl-sensitive restriction enzyme-qPCR (MSRE-qPCR) on genomic DNA in a subset of *DRB1*15:01* heterozygous individuals (haplotype-specific primers were adapted from Olerup et al.[26], see Methods and Supplementary Data 3). The *DRB1*15:01* allele exhibited a lower cg06032479 methylation compared to other tested alleles ($p = 7.9 \times 10^{-3}$ and $p = 2.2 \times 10^{-3}$ compared to *DRB1*03* and *DRB1*04*, respectively, Mann–Whitney test, Fig. 2e, the individuals' data are shown in Supplementary Fig. 1a). Taken together, these data demonstrate specific hypomethylation of the *DRB1*15:01* allele.

We next investigated potential functional consequences of differential methylation at *HLA-DRB1* in the same monocyte cohort. To minimize any sequence-bias in amplification, expression was quantified using four sets of primers mapping to less polymorphic segments of the transcript (Supplementary Data 3). *HLA-DRB1* expression showed a strong negative correlation with methylation at CpG sites in the region encompassing exon 2 (Fig. 3a) regardless of the primer pair used. Accordingly, *DRB1*15:01* carriers displayed significantly higher expression of *HLA-DRB1* in monocytes compared to non-carriers ($p = 1.1 \times 10^{-5}$ and $p = 7.5 \times 10^{-4}$ for homozygotes and heterozygotes vs. non-carriers, respectively, ANOVA with Dunn's multiple comparison test, $n = 58$, Fig. 3b). This suggests a functional link between low DNA methylation and high *HLA-DRB1* expression, which was further supported by the increase in *HLA-DRB1* expression in peripheral blood mononuclear cells (PBMCs) treated with the demethylating drug 5-Aza-2′-deoxycytidine ($p = 0.036$, ANOVA with Turkey's multiple comparison test, Fig. 3c). Using allele-specific primers, we quantified the expression of each *HLA-DRB1* allele carried by individuals heterozygous for *DRB1*15:01*. Compared to all other alleles, *DRB1*15:01* displayed significantly lower methylation ($p = 3.9 \times 10^{-9}$, Mann–Whitney test, Fig. 3d, Supplementary Fig. 1a) and higher expression ($p = 5.4 \times 10^{-7}$, *t*-test, Fig. 3e, Supplementary Fig. 1b). Similar results were obtained using an allele-specific SNP analysis approach (see Methods, Supplementary Fig. 1b). The allele-specific analysis in monocytes from the same individuals showed a negative correlation between methylation and expression (Spearman $r = -0.56$, $p = 1.7 \times 10^{-3}$, Supplementary Fig. 1c). As expected, global *HLA-DRB1* levels correlated with the sum of both alleles (Supplementary Fig. 1e), indicating that, for the majority of tested haplotypes, high *HLA-DRB1* expression in monocytes reflects overexpression of the *DRB1*15:01* allele.

We tested whether methylation in the region encompassing exon 2 of *DRB1*15:01* can exert regulatory properties on gene expression. We addressed enhancer and promoter activity and the effect of DNA methylation levels of the region using methylation-sensitive CpG-free vector-based reporter systems. The inserts were methylated using two different methyltransferases, *SssI* and *HhaI*, which methylate all CpG sites (50 CpGs) or only the internal cytosine residue in GCGC (5 CpGs), respectively. Notably, the region encompassing exon 2 of *DRB1*15:01* displayed significantly decreased enhancer ($p = 1.3 \times 10^{-4}$) and promoter ($p = 9.7 \times 10^{-3}$) activity if the insert was fully methylated (ANOVA with Turkey's multiple comparison test, Fig. 3f, g, Supplementary Fig. 1f). This indicates that DNA methylation is a regulatory feature of the region encompassing exon 2 of *DRB1*15:01* and that the hypomethylated *DRB1*15:01* has a capacity to drive higher gene expression.

Although the size of the monocyte cohort was insufficient for formal methylation mediation analysis, our data suggest that hypomethylation and predominant expression of *DRB1*15:01* could be a mechanism by which *DRB1*15:01* confers risk of MS.

**Genome-wide analysis of methylation mediation.** Our findings in monocytes suggest that DNA methylation may be an intermediary of genetic risk in MS. Therefore, we set out to identify the epigenetic marks that may mediate the genetic risk for MS by integrating genome-wide genetic and epigenetic analysis, similar to the original use of the Causal Inference Test (CIT) method in our study on RA[19] (Fig. 1). This method is robust to issues such as pleiotropy and reverse confounding that are likely to occur in complex diseases. Using whole blood samples from an independent genotyped (500 K) MS case-control cohort ($n = 279$, cohort 2, Supplementary Data 1), analyzed with 450K arrays, we applied the CIT method[19] with genotype as a causal factor, DNA methylation as a potential mediator and MS as the phenotypic outcome (G, M, and Y, respectively, Fig. 4a). As regulatory methylation changes generally encompass multiple CpGs, and to minimize potential measurement errors, we sought to identify DMRs[27]. We first identified seven DMRs that associated with MS after adjustment for confounders (FWER < 0.05), all of which mapped to the *HLA* class II region (*HLA-DRB5, -DRB1, -DQA1, -DQB1* genes, Table 1). The methylation levels within these seven DMRs were correlated in MS cases and healthy controls (Supplementary Fig. 2), forming genetically controlled methylation clusters known as GeMes[17]. We found that these DMRs were under the genetic control of 202 unique SNPs, with 875 significant SNP-DMR pairs (Bonferroni-adjusted $p < 0.05$), also known as methylation Quantitative Trait Loci (meQTLs). Out of the 202 SNPs, 52 were significantly associated with MS status (adjusted $p < 0.05$, maxT permutation), and all were located in the *HLA* region. We finally performed CIT analysis to identify the genetic variants that are independent of case/control status after adjusting for DNA methylation, suggesting mediation (Fig. 4a). Among the 52 SNPs, 50 were significant after causal analysis

**Fig. 3** *DRB1*15:01*-associated expression in monocytes. **a** Spearman correlation between DNA methylation at *HLA-DRB1* (exon 2) (450K arrays) and *HLA-DRB1* expression quantified by RT-qPCR using primers targeting different segment of the transcript (exon 6 by primer sets 1 and 2, exon 1 by primer set 3, and exon 4-6 by primer set 4). **b** *HLA-DRB1* expression according to *DRB1*15:01* haplotype: homozygous (+/+, $n = 7$, red), heterozygous (+/−, $n = 21$, orange), and non-carriers (−/−, $n = 30$, blue) quantified by RT-qPCR (using primer set 1). **c** *HLA-DRB1* expression ($n = 4$ independent experiments) in PBMCs from *DRB1*15:01* non-carriers treated with 5-Aza-2′-deoxycytidine (5-aza-dC). **d** *HLA-DRB1* allele-specific methylation of cg06032474 quantified by MSRE-qPCR from *DRB1*15:01* heterozygote (+/−) individuals ($n = 20$, red and blue colors representing *15:01* and the other allele, respectively). **e** Relative expression of each *HLA-DRB1* allele from *DRB1*15:01* heterozygote (+/−) individuals quantified by allele-specific qPCR ($n = 26$, red and blue colors representing *15:01* and the other allele, respectively). Enhancer (**f**) and promoter (**g**) activity of the exon 2 region of *DRB1*15:01* using CpG-free promoter-containing (Lucia) and promoter-free (SEAP) reporter gene vectors, respectively. Constructs were partially or fully methylated using HhaI and SssI enzymatic treatment, respectively. Results show relative activity (Lucia or SEAP normalized against Renilla) using five replicates in a representative experiment performed at least 2–3 times. Efficiency of the in vitro methylation is shown in Supplementary Fig. 1f. NA not applicable due to absence of promoter. **b**–**g** Data are presented as Tukey boxplots; *$p < 0.05$ **$p < 0.01$, ***$p < 0.001$, using Spearman test (**a**), ANOVA with Dunn's (**b**), or Turkey's multiple comparison tests (**c**, **f**, **g**), the Mann–Whitney test (pooled alleles A vs. pooled alleles B) (**d**), and *t*-test (pooled alleles A vs. pooled alleles B) (**e**)

using CIT ($p < 0.05$) including all seven DMRs (Table 1, Fig. 4b) and thus represent potential methylation-mediated relationships between SNPs and MS disease risk. The type of disease or treatment status at sampling had no effect on the methylation levels at any of the seven DMRs ($p > 0.3$ for all comparisons). Results of the analysis are given in Supplementary Data 4 and one example (chr6: 32552039-32552350; rs3135338) is illustrated in Fig. 4c. We further replicated the methylation differences between MS cases and controls for six out of the seven DMRs using sorted

CD14$^+$ monocytes ($n = 36$, cohort 1), CD19$^+$ B cells ($n = 29$), as well as CD4$^+$ ($n = 33$) and CD8$^+$ ($n = 29$) T cells, with the least pronounced differences being observed in CD8$^+$ T cells (Fig. 4c, Supplementary Data 5). Collectively, our data suggest a functional link between DNA methylation at the *HLA* class II locus and the risk of developing MS.

Remarkably, four of the seven DMRs reside within the *HLA-DRB1* gene and the two largest (DMR3 and DMR4) encompass the same CpGs that we have identified as associated with *HLA-*

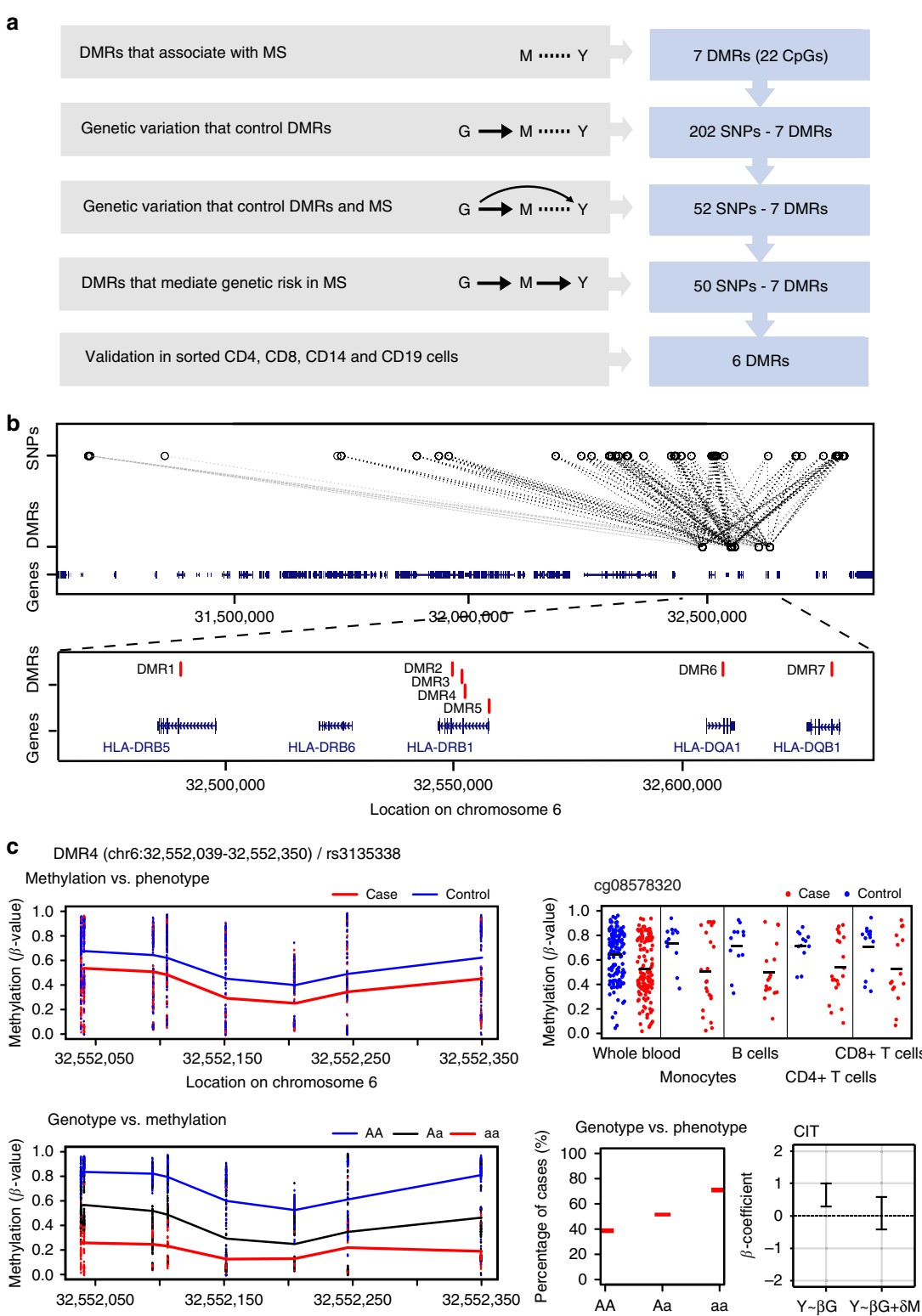

*DRB1* expression in monocytes. We addressed a potential functional impact of the identified methylation-mediated SNPs on the expression of *HLA-DRB1* using RNA-seq data from PBMCs[28] (cohort 3, $n = 156$, Supplementary Data 1). We found that the risk alleles of all genotyped SNPs ($n = 34$) were associated with high *HLA-DRB1* expression in PBMCs and the risk alleles for 28 SNPs were also associated with low methylation at DMR3 or DMR4 in blood (Supplementary Data 6). This suggested a causal link between the identified SNPs, DNA methylation at *HLA-DRB1* exon 2 and *HLA-DRB1* expression. To address this, we employed a two-sample Mendelian randomization (MR) using the SNPs previously identified as meQTLs for DMR3 or DMR4 in the blood cohort and eQTLs for *HLA-DRB1* in the PBMC cohort. We utilized the Egger's regression slope ($\beta_{MR}$)[29] to estimate the causal effect (see Methods). The MR-Egger analysis revealed a significant causal relationship between DNA methylation at DMR3 ($\beta_{MR} \pm SE = -1.92 \pm 0.33$, $p = 6.75 \times 10^{-9}$) or DMR4 ($\beta_{MR} \pm SE = -1.80 \pm 0.30$, $p = 3.04 \times 10^{-9}$) and *HLA-DRB1* gene expression (Fig. 5 and Supplementary Data 6). We also verified that the inferred causal direction is correct using the MR Steiger test for directionality[30]. Namely, for the variant with the strongest meQTL and eQTL effect (rs3132946), DNA methylation was found to cause changes in *HLA-DRB1* expression ($p = 2.87 \times 10^{-3}$ and $p = 8.65 \times 10^{-4}$ for DMR3 and DMR4, respectively, Supplementary Fig. 3).

These findings provide additional insight into the molecular mechanisms of variants in the *HLA* region that likely mediate the risk for developing MS through changes in DNA methylation and expression of *HLA-DRB1*.

**Methylation mediation of *DRB1\*15:01* and a protective variant**. Considering the complex structure of the *HLA* locus, with LD extending over large distances, and our findings of *DRB1\*15:01*-specific hypomethylation in monocytes, we investigated whether the identified methylation-mediated SNPs confer risk of MS independently of the *DRB1\*15:01* haplotype (Fig. 1). We addressed this question in a large Scandinavian case-control cohort ($n = 8172$ cases and 13,263 controls, cohort 4, Supplementary Data 1) by testing association of the SNPs (45/50 with genotype data) with MS after adjusting for *DRB1\*15:01* (and associated terms) and all other established risk variants in the *HLA* locus[5] (Fig. 6a, Supplementary Data 7). The majority of methylation-mediated SNPs (41/45) showed limited evidence of association with MS ($p \geq 1 \times 10^{-5}$) after conditioning on *DRB1\*15:01* (Fig. 6a), suggesting that they confer risk from *DRB1\*15:01*. However, three methylation-mediated SNPs displayed suggestive association with MS ($p < 1 \times 10^{-5}$) after adjusting for all known MS-associated variants in the *HLA* locus[5] (Fig. 6a, Supplementary Data 7), suggesting potential novel variants. We followed up these associations by performing a meta-analysis using the Scandinavian cohort and three additional MS case-control cohorts (Supplementary Data 1) from Sweden ($n =$

1418 cases and 1058 controls), Germany ($n = 3934$ cases and 8455 controls)[9], and Iceland ($n = 735$ cases and 148,571 controls), adjusting for all known MS-associated variants in the *HLA* locus[5] (Supplementary Data 7). rs9267649 displayed a genome-wide significant association with MS ($p = 3.32 \times 10^{-8}$, OR = 0.86) with a similar protective effect in all cohorts (Fig. 6b). Three additional methylation-mediated SNPs (rs2227956, rs2395182, and rs9271640) exhibited suggestive association with MS ($p < 1 \times 10^{-5}$) with rs2395182 and rs9271640 being in LD ($r^2 = 0.72$, Supplementary Fig. 4). Among these suggestive SNPs, only rs2227956 ($p = 7.08 \times 10^{-8}$, OR = 0.86) was in high LD with rs9267649 ($r^2 = 0.96$, Supplementary Fig. 4), thus representing the same association. Interestingly, the protective alleles at both SNPs were associated with higher blood methylation levels at DMR3 (chr6: 32551749-32551949) in exon 2 of the *HLA-DRB1* gene and a lower *HLA-DRB1* gene expression in PBMCs, particularly in *DRB1\*15:01* heterozygous individuals (Fig. 6c, d, Supplementary Fig. 5). This suggests that the protective variant could be counteracting the effect of the major *DRB1\*15:01* allele at the molecular level.

Collectively, our findings strongly suggest that DNA methylation mediates the risk of *DRB1\*15:01* on MS. Moreover, we identify an MS-associated variant which was not detectable by conventional genetic studies and protects against MS potentially via modulating DNA methylation at *HLA-DRB1*.

## Discussion

Over the past decade, series of GWAS and custom-designed arrays have steadily added more regions to the list of MS-associated loci[4,7–10]. However, identification and interpretation of the causal variants remains difficult and their mechanisms are still largely unknown. Considering that the majority of susceptibility loci reside in regulatory regions of the genome and the important role of DNA methylation in gene regulation, we sought to investigate DNA methylation in MS patients in the context of the underlying genetic variation. Our data strongly suggest that DNA methylation in the *HLA* class II locus, especially encompassing exon 2 of the *HLA-DRB1* gene, mediates the effect of *DRB1\*15:01* and of a protective *HLA* variant, which has not been previously reported, on *HLA-DRB1* expression and the risk of MS. These findings provide new insights into the molecular mechanisms of MS susceptibility and suggest alternative therapeutic strategies based on modulating *HLA-DRB1* levels.

Although the *HLA* region has been known as the strongest genetic risk factor for MS for over 40 years, the exact causal gene(s) and the mechanisms by which they affect MS susceptibility are still elusive. It is generally accepted that disease-associated variants in *HLA-DRB1* primarily influence the structure of the peptide-binding groove encoded by exon 2. Altered amino acid residues of the HLA class II beta chain expressed on the APCs could lead to a changed T cell repertoire that causes autoimmune responses in the CNS[31–34]. However, specific antigens in MS have

**Fig. 4** Genotype-dependent candidate DMRs that mediate genetic risk in multiple sclerosis (MS). **a** Summary workflow and results for identifying epigenetically mediated genetic risk factors for MS. The diagrams on the right represent the relationships between genotype (G), DNA methylation (M), and MS (phenotype, Y). Dashed lines, the association relationship; arrows, the causal relationship. **b** Association between candidate genetic risk-mediating DMRs and genotype. Each dashed line represents a potential mediation relationship between an SNP and a DMR as determined by the CIT. **c** Association between DNA methylation levels at DMR4 chr6:32552039-32552350 (located in exon 2 of *HLA-DRB1*) that mediates genetic risk in MS and phenotype (top panels), with red and blue colors representing cases and controls, respectively, in blood cells ($n = 279$) (top left), and in sorted CD14+ monocytes ($n = 36$), CD19+ B cells ($n = 29$), CD4+ ($n = 33$), and CD8+ ($n = 29$) T cells for cg08578320 (top right), or genotype rs3135338 (bottom left panel) with blue, black, and red colors denoting AA, Aa, and aa genotypes. Bottom right panels: association between genotype (rs3135338) and phenotype and CIT. Red horizontal bars mark percentage of cases for each genotype. Coefficient ($\beta$) represents the dependence of the MS phenotype on genotype, with or without adjusting for DNA methylation. The error bars represent the 95% confidence interval for the coefficient $\beta$. In the case of the methylation-mediated model, the absolute value of the observed G:Y relationship strength reduces toward zero when adjusting for methylation. DMR: differentially methylated region, SNP: single nucleotide polymorphism, CIT: causal inference test. The full list of SNP-DMR pairs is shown in Supplementary Data 4

**Table 1 DMRs that mediate genetic risk in multiple sclerosis**

| Probe | DMR | Location | Nber SNPs | FWER[a] | Value[a] | Gene | Description | TF[b] | Chromatin state[b] |
|---|---|---|---|---|---|---|---|---|---|
| cg26981746 cg12015991 | DMR 1 | chr6:32490012-32490043 | 47 | 0.013 | 0.12 | HLA-DRB5 | Intron 1/2 | EZH2, ZNF263 | Weak transcribed |
| cg13910785 cg23905789 | DMR 2 | chr6:32549849-32549935 | 41 | 0 | 0.21 | HLA-DRB1 | Intron 2/3 | POLR2A | Transcriptional elongation |
| cg11404906 cg19575208 cg08845336 cg15568074 | DMR 3 | chr6:32551749-32551949 | 41 | 0 | −0.14 | HLA-DRB1 | Intron 2/3-Exon 2 | POLR2A, ZNF263, SIN3A, CTCF | Weak/poised promoter, weak/poised enhancer |
| cg08578320 cg09139047 cg15602423 cg15982117 cg16514085 cg14645244 cg00211215 cg09949906 | DMR 4 | chr6:32552039-32552350 | 42 | 0 | −0.16 | HLA-DRB1 | Exon 2-Intron 1/2 | POLR2A, ZNF263, SIN3A, CTCF | Weak/poised promoter, weak/poised enhancer |
| cg24760581 cg10385522 | DMR 5 | chr6:32557970-32558175 | 27 | 0 | 0.18 | HLA-DRB1 | TSS1500 | YY1 | Strong enhancer |
| cg24470466 cg17421046 | DMR 6 | chr6:32608858-32608879 | 19 | 0.005 | 0.13 | HLA-DQA1 | Intron 1/2 | POLR2A, CHD1 | Weak promoter |
| cg13423887 cg05341252 | DMR 7 | chr6:32632694-32632715 | 43 | 0.004 | −0.13 | HLA-DQB1 | Exon 2 | POLR2A, EZH2, CHD1, CTCF | Active/poised promoter |

DMR: differentially methylated region, chr: chromosome, Nber: number, SNP: single nucleotide polymorphism, FWER: family wise error rate, TSS: transcription starting site, TF: transcription factor
[a]Methylation vs. phenotype
[b]From ENCODE

yet to be defined[35], which hinders the elucidation of the mechanisms underlying susceptibility conferred by the major risk haplotype DRB1*15:01, as well as development of antigen-specific therapies. On the other hand, it appears that the structural theory alone does not fully explain the association with MS, and recent studies have suggested that association of DRB1*15:01 with MS might also be related to gene expression[28,36–38]. Consistent with this, we found that in monocytes of DRB1*15:01 carriers, HLA-DRB1 is unmethylated and expressed at a higher level compared to other haplotypes. We then utilized causal inference strategies to test the hypothesis that DNA methylation mediates the effect on HLA-DRB1 expression and, in turn, on MS susceptibility.

We first applied CIT in a case-control GWAS cohort and found statistical evidence of DNA methylation at HLA-DRB1 mediating the risk for MS from several SNPs located in the extended HLA locus. The majority of these SNPs conferred the effect from DRB1*15:01, as demonstrated by the lack of association after conditioning on DRB1*15:01. The impact of the DRB1*15:01 genotype on HLA-DRB1 methylation was similar in MS patients and healthy controls (Supplementary Fig. 5). This strong genetic effect can explain the observed methylation differences between MS patients and controls in all studied cell types, i.e., CD14+ monocytes, CD19+ B cells, CD4+ T cells, and CD8+ T cells. Similar to our findings, HLA-DRB1 methylation differences between MS patients and controls have been reported in CD4+ T cells and suggested to be partially dependent on DRB1*15:01[12]. Previous meQTLs studies have reported strong genetic regulation of methylation at CpGs shared by the DMRs identified in our study in white blood cells[21], pancreatic islets[20], and brain samples[39]. Since these studies have not focused on DRB1*15:01, we have investigated the SNPs underlying the reported meQTLs and we found that many are in high LD with DRB1*15:01 and show the same direction of effect as in our study. It is thus not surprising that HLA-DRB1 methylation differences between cases and controls exist in multiple cells types, although the functional consequences of this genotype-driven methylation likely differ between distinct cell types and depend on the

contribution of HLA class II molecules to a particular cell type-specific function.

Most of the CpGs identified by CIT mapped to the same DMRs in exon 2 of HLA-DRB1 found in monocytes, supporting a functional link between genetic variation, methylation, expression and risk of MS. Since DNA methylation changes can actively impact gene expression or be a consequence of transcriptional activity in the locus[40], we further addressed the potential causal effect of DNA methylation in exon 2 of HLA-DRB1 on expression both experimentally and analytically. Using an in vitro reporter system, we demonstrated that the exon 2 sequence of DRB1*15:01 exerts regulatory properties on gene expression in a DNA methylation-dependent manner. This suggests that hypomethylation of exon 2 could mediate the effect of DRB1*15:01 on HLA-DRB1 gene expression. This is also supported by the significant causal relationship between methylation at DMRs in exon 2 and HLA-DRB1 expression in PBMCs, obtained using two-sample MR. The directionality of the causal relationship was further confirmed using a method that combines MR with a Steiger test (MR-Steiger)[30]. Altogether, our data strongly support that DRB1*15:01-dependent and DNA methylation-mediated levels of expression, together with the structural characteristics of the DRB1*15:01 molecule, contribute to MS risk. Indeed, both the quantity and the quality of the peptide-HLA complexes define immune responses[41–43] and levels of the HLA genes in humanized HLA and TCR transgenic mice influence the severity of experimental MS-like disease[44]. Our findings highlight the potential of using alternative or complementing strategies to antigen-specific therapies in MS, which would reduce the expression of HLA-DRB1. This strategy might be relevant beyond MS, given the associations between genetic variants and DNA methylation in the HLA class II region in several immune-mediated diseases[19–21].

The CpGs in exon 2 of HLA-DRB1 that mediate the effect on MS risk belong to an intragenic CpG island that shows enrichment for binding sites of the 'architectural' regulatory protein of the genome CCCTC-binding factor (CTCF) (Table 1). A DNA

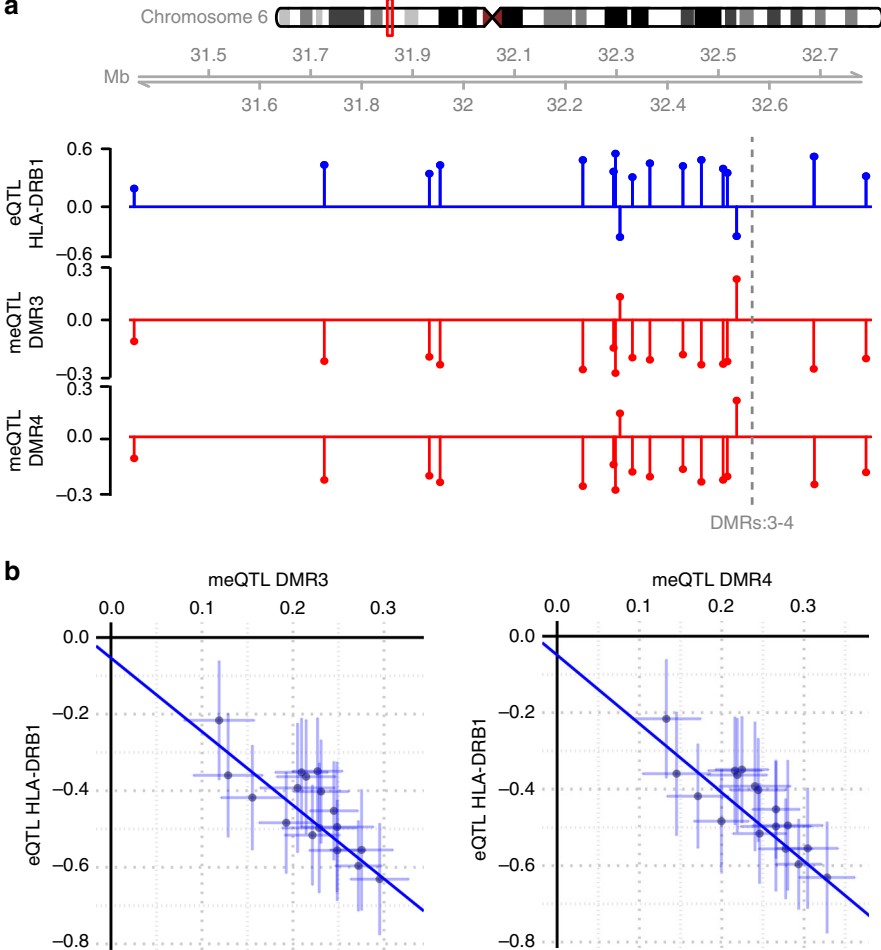

**Fig. 5** Two-sample Mendelian randomization (MR). **a** The top plot shows the effect size from the eQTL analysis (outcome, $n = 156$, blue) in the PBMC cohort for the SNPs included in the MR. The middle and bottom plots (red) show the effect sizes from the meQTL analysis (exposure, $n = 279$, red) in the blood cohort for DMR3 and DMR4, respectively, for the SNPs included in the MR. The location of two regions is marked with a dashed line. Further details are given in Methods. **b** Scatterplots of the effect sizes for meQTL (x-axis) and eQTL (y-axis) for DMR3 (left) and DMR4 (right). The effect size and the 95% confidence intervals are shown. The blue line represents the causal estimate using the Egger regression. All the associations with the exposures were set to be positive and the associations with the outcome were re-oriented

methylation-dependent role of intergenic CTCF has previously been suggested in the gene-specific regulation of the *HLA* class II locus[45,46]. Furthermore, an insulator function of intragenic CTCF has been reported when bound to intron 2 of *DRB1*04*[47]. In line with this, non-canonical intragenic CTCF occupancy has been shown to regulate intragenic chromatin boundaries, ultimately affecting gene-specific transcription[48,49] and alternative splicing[50,51]. However, whether differential methylation at this locus affects binding of regulatory proteins and fine-tunes regulation of specific genes within the *HLA* locus during disease development is yet to be explored.

In addition to the methylation-mediated SNPs that conferred the effect from *DRB1*15:01*, we found potential associations with MS that were independent of any known MS variant in the *HLA* region[5]. A previously unreported protective variant was discovered through significant and highly suggestive association of two SNPs in high LD ($r^2 = 0.96$), rs9267649 and rs2227956. Two additional suggestive SNPs in high LD with each other ($r^2 = 0.72$), rs2395182 and rs9271640, were associated with the risk of developing MS. The two latter SNPs do not necessarily represent a variant that is independent of the significant rs9267649 protective variant, as there is correlation between the SNPs ($r^2 = 0.23$ and $r^2 = 0.22$ for rs2395182 and rs9271640, respectively). In contrast to the potent hypomethylation of *DRB1*15:01*, the

protective variant was associated with increased DNA methylation of DMR3 in exon 2 of *HLA-DRB1* and with lower *HLA-DRB1* expression in PBMCs. Due to the strong influence of *DRB1*15:01* on methylation and expression levels and its correlation with rs9267649 ($r^2 = 0.40$), the significant functional impact of the protective variant could only be observed in individuals stratified by *DRB1*15:01* genotype, and it was particularly evident in heterozygotes. This opposing effect suggests a putative interaction with *DRB1*15:01* and its potential to alter, possibly antagonize, the effect of the major risk allele at the molecular level. Interactions within the *HLA* region have previously been suggested in immune-mediated diseases[5,52]. The protective variant might influence *HLA-DRB1* transcription through long-range interactions between regulatory regions, as previously suggested to occur within the *HLA* class II locus[53]. However, given far-extending LD in the region, it is also plausible that the associated SNP tags a true causal variant elsewhere in the *HLA* locus, warranting future efforts to refine this association.

Our findings highlight the importance of integrating genetic and epigenetic data to explore molecular mechanisms underlying causal variants and to identify variants that might escape detection by conventional genetic studies. Given the robust genetic association of the *HLA* region with susceptibility to immune-mediated diseases, our findings, together with other studies,

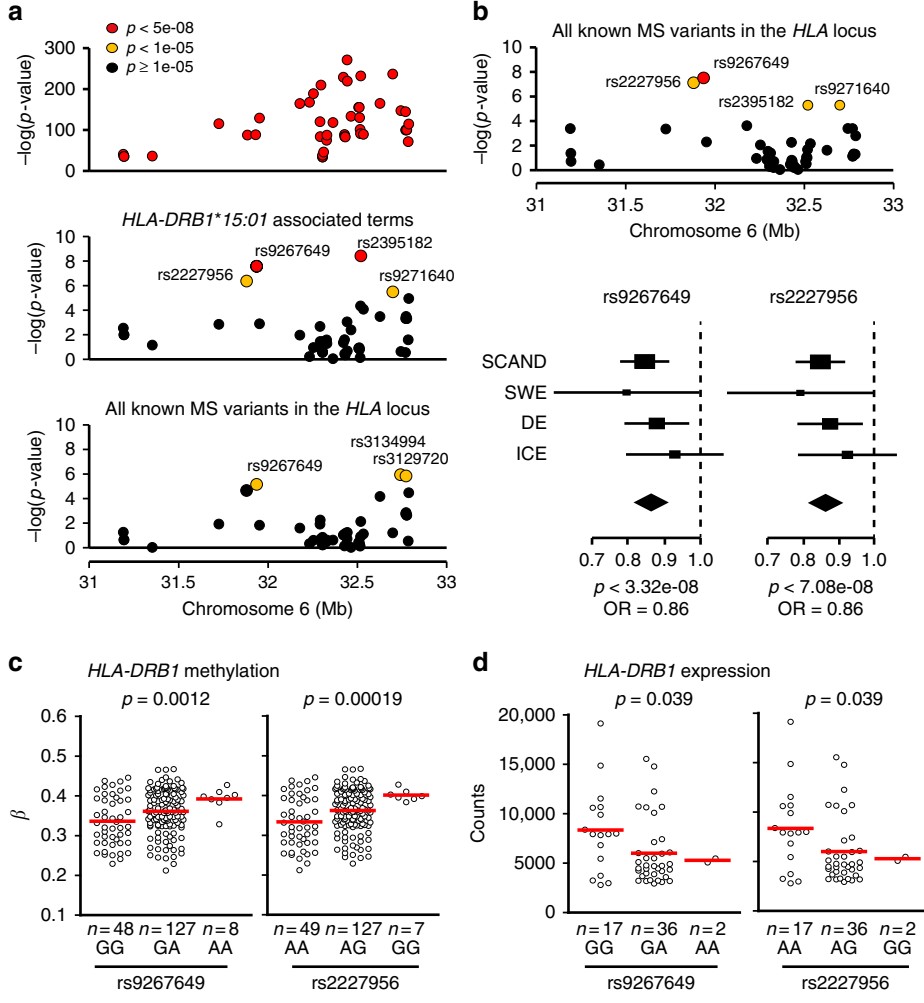

**Fig. 6** Association of methylation-mediated SNPs with multiple sclerosis (MS). **a** Association between SNPs and MS in the Scandinavian cohort (SCAND, 8172 cases and 13,263 controls) after adjustment for four PCAs (upper panel), the *DRB1*15:01* associated terms (middle panel) and all 13 established MS risk variants in the *HLA* locus[5] (lower panel) (for details see Methods). **b** Association between SNPs and MS based on meta-analysis of SCAND and three additional cohorts from Sweden (SWE), Germany (DE), and Iceland (ICE) (upper panel, 14,259 cases and 171,347 controls) and Forest plots (lower panel) representing odds ratios (OR, square, proportional to weight) and associated confidence intervals for each cohort and the summary measure (diamond) for the significant (rs9267649) and suggestive (rs2227956) SNPs, with dotted vertical line of no effect. **a, b** The $-\log_{10}(p$-value) of 47 out of 50 SNPs and their position on chromosome 6 are given on the *y*- and *x*-axis, respectively. Colors of circles correspond to different thresholds of statistical significance, red: $p$-value $< 5 \times 10^{-8}$ (genome-wide significance), orange: $1 \times 10^{-5} < p$-value $< 5 \times 10^{-8}$ (suggestive significance) and black: $p$-value $\geq 1 \times 10^{-5}$ (non-significant). **c** Methylation values at DMR3 (exon 2) of *HLA-DRB1* gene in *DRB1*15:01* heterozygous MS patients and healthy controls ($n = 183$) stratified according to the rs9267649 (left panel) and rs2227956 (right panel) genotype. **d** *HLA-DRB1* gene expression in *DRB1*15:01* heterozygous MS patients and non-MS controls ($n = 55$) stratified for the rs9267649 (left panel) and rs2227956 (right panel) genotype. **c, d** Significance was estimated using linear regression (for details see Methods)

suggest a role for DNA methylation in the pathogenesis of MS and autoimmune diseases in general. This in turn opens new avenues for development of therapeutic strategies aiming at controlling immune reactions by modulating HLA protein levels.

## Methods

**Cohorts**. Description of the cohorts is shown in the Supplementary Data 1. Briefly, all samples used for DNA methylation and expression studies in MS, i.e., peripheral blood cells and sorted cells from blood, were collected in Sweden between 2005 and 2011. No formal sample size calculation was conducted, all available samples that passed quality control have been included in all analyses. Cohort 1 consisted of monocytes isolated from 62 MS patients and 20 healthy controls and was used for genome-wide and locus-specific DNA methylation and expression analyses. Thirty-six of them were selected for genome-wide analysis based on the sufficient amount of DNA required for 450K arrays and the matching clinical parameters (sex and age) between the groups. Of the 23 MS patients, 91% (21/23) were not treated at the time of sampling (either never treated or they stopped treatment at least 6 months before). All samples with DNA of sufficient quantity ($n = 49$) were used

for pyrosequencing validation. All samples with available RNA of sufficient quality ($n = 58$) were used for qPCR-based gene expression analysis. All samples that were *DRB1*15:01* heterozygotes and where DNA/RNA of a sufficient quantity and quality was available, were used for allele-specific methylation and expression analyses. Details of individuals used in specific analyses are provided in the Supplementary Data 1. Cohort 2 used for CIT analysis on peripheral blood cells included 140 MS patients with relapsing-remitting (RRMS, $n = 121$), primary progressive (PPMS, $n = 4$), or secondary progressive (SPMS, $n = 15$) disease, and 139 healthy controls. In total, 65% (91/140) of MS patients were treated at the time of sampling but the majority, 85% (77/91), received drugs that have a moderate impact on disease activity (e.g., 63 received interferon beta preparations and 12 Glatiramer acetate). Cohort 3 used for expression QTL analysis comprised PBMCs from 156 patients, including 21 clinically isolated syndrome (CIS), 105 MS patients, and 30 non-inflammatory neurological disease controls (NINDC), such as neuralgia, paresthesia, sensory symptoms, vertigo, tension headache[28]. Cohort 4 comprised four non-overlapping case-control cohorts used for conditional association studies. The Scandinavian (SCAND) cohort consisted of 8172 MS cases and 13,263 controls from the three Swedish national studies of MS; EIMS[54], GEMS[55], and IMSE[56]. In addition, cases were included from a local biobank of MS cases as well as a cohort of Swedish blood donors[57] and Swedish controls from the OLIVIA[58],

SASBAC[59], and Twingene[60] studies. Danish cases and Norwegian cases and controls from in the IMSGC WTCCC2 GWAs were also included in the SCAND cohort[4]. The Swedish (SWE) cohort consisted of additional MS cases ($n = 1418$) and controls ($n = 1058$) from Sweden. The German (DE) cohort comprised MS cases ($n = 3934$) recruited from multiple sites in Germany[9] and matched controls ($n = 8455$) from several population-based cohorts across Germany namely KORA[61], HNR[62], SHIP[63], DOGS[64], and FoCus[65], as previously described[9]. The Icelandic (ICE) cohort consisted of 735 MS cases diagnosed between 1950 and 2005 and followed up at Landspítali, the National University Hospital of Iceland[10] and 148,571 population-based controls available through on-going projects at deCODE genetics. The study was approved by the National Bioethics Committee of Iceland (VSN_15-212) and conducted in agreement with conditions issued by the Icelandic Data Protection Authority (DPA). All subjects who donated DNA samples signed informed consent. Personal identifiers of the patient data and biological samples were encrypted by a third party system approved and monitored by the DPA. An additional 333 healthy individuals with genotype and 450K array data have been obtained from the EIRA (the Epidemiological Investigation of Rheumatoid Arthritis) cohort from Sweden, GSE42861[19]. All study participants had given their informed consent according to ethical board approval. Specifically, the study complies with ethical regulations and was approved by the Regional Ethical Board at Karolinska Institutet (Solna, Sweden), Copenhagen and Fredriksberg, (Denmark), Oslo (Norway) and Icelandic national bioethics committee (VSN 15-212). For the German cohorts, the local ethics committees of the individual institutions contributing patients provided positive votes for each local study (Germany): Technical University Munich, Max Planck Institute of Psychiatry Munich (approved by the ethics committee of the Medical Faculty at the Ludwig Maximilians University, Munich), University of Münster, University Medical Center of the Johannes Gutenberg University Mainz (approved by the ethics committee of the medical association of Rheinland-Pfalz with the approval ID 837.019.10(7028)), Ruhr-Universität Bochum (reg. nr. 4319 12), University Medical Center Hamburg-Eppendorf (approved by the ethics committee of the Ärztekammer Hamburg), Charité—Universitätsmedizin Berlin, University of Rostock, University of Heidelberg, University of Marburg, and University of Leipzig.

**Sample preparation.** PBMCs were obtained from 50 ml of blood and extracted using standard Ficoll gradient procedures directly after collection. Monocytes were isolated using CD14$^+$ positive selection on MACS microbeads magnetic separation (Miltenyi), according to manufacturer's instructions (>95% purity). Sorting of the CD4$^+$ and CD8$^+$ T cell and CD19$^+$ B cell populations was performed from the negative fraction obtained after sorting of monocytes by adding fluorochrome conjugated antibodies against human CD4 (clone SK3, APC-conjugated, Becton Dickinson), CD8 (clone SK1, FITC-conjugated, Becton Dickinson), CD3 (clone UCHT1, PE-conjugated, BD Bioscience), and CD19 (clone SJ25C1, APC-Cy7-conjugated, Becton Dickinson) using high speed MoFlo™ cell sorter with >99% purity (Beckman Coulter, Inc). Extraction of genomic DNA and RNA from sorted cells was performed using Gen Elute Mammalian Genomic DNA Miniprep kit (Sigma-Aldrich) and RNeasy Mini kit (Qiagen), respectively. The amount and quality of DNA/RNA was accessed with a NanoDrop ND-1000 Spectrophotometer (NanoDrop Technologies Inc). Processing of samples for 450K arrays was done at BEA core facility, Karolinska Institutet (Stockholm) for monocytes and CD4$^+$ T cells, and at Johns Hopkins University School of Medicine (Baltimore) for whole blood, CD8$^+$ T cells and CD19$^+$ B cells. For each experiment, MS cases and controls were randomized and technicians performing the genotyping, RNA-seq and 450K arrays were blinded to the MS disease status during the experiments.

For ex vivo PBMCs culture, PBMCs from non-carrier individuals of DRB1*15:01 risk haplotype were cultured in Roswell Park Memorial Institute (RPMI) medium supplemented with 10% fetal bovine serum, 1% penicillin/streptomycin, 2 mM L-Glutamine, and 1 mM sodium pyruvate at 37 °C in 5% CO$_2$. Cells were stimulated with phorbol 12-myristate 13-acetate (PMA, Sigma) and exposed to different doses (0.5 μM, 5 μM) of 5-Aza-2′-deoxycytidine (Sigma) for 3 days prior to harvesting and subsequent RNA/DNA extraction using AllPrep DNA/RNA Micro Kit (Qiagen).

**Expression analyses of HLA-DRB1 in monocytes.** The full list of primer sequences used in this study is shown in Supplementary Data 3. Expression of HLA-DRB1 was quantified using primers targeting different segments (exon 1, exon 4–6, and exon 6) of the transcript. Allele-specific expression analysis of HLA-DRB1 variants was assessed using primer sets mapping to exon 2 of the transcript and specific for each group of alleles (targeting multiple haplotype-specific SNPs, adapted from Olerup et al.[26] and confirmed using in silico alignment to IPD-IMGT/HLA database). Similar allele-dependent expression results were obtained by applying a single SNP-based approach targeting rs9270303 using two allele-specific forward primers specific for either DRB1*15:01/*01 or the other HLA-DRB1 alleles (*03–*14), and one common reverse primer. Real-time PCR was performed on a BioRad CFX384/C1000 Real-Time Detection System with a three-step PCR protocol using SYBR green fluorophore: 95 °C:3 min, followed by 40 cycles of 95 °C:10 s, 60 °C (65 °C for the allele-specific SNP-based amplification): 30 s and 72 °C:30 s. Relative quantification of mRNA levels was performed using the standard curve method, with amplification of target mRNA and endogenous control Actin mRNA. Assumption of equal variance between groups was tested using Brown-Forsythe test. Differences in HLA-DRB1 levels between DRB1*15:01 homozygotes, heterozygotes and non-carriers, and after of 5-Aza-2′-deoxycytidine were tested using ANOVA with Dunn's and Turkey's multiple comparison tests, respectively. DRB1*15:01 expression was compared to expression of the other allele in DRB1*15:01 heterozygotes using t-test. Correlation between HLA-DRB1 levels and methylation at HLA-DRB1-associated CpGs generated with 450K arrays was performed using the Spearman test. All statistical analyses were performed in GraphPad Prism 6 and 7 (GraphPad Software).

**Methylation analyses of HLA-DRB1 in monocytes.** For pyrosequencing and cloning-sequencing, 500 ng of genomic DNA from each sample was bisulfite-converted using an EZ DNA methylation Kit (ZYMO research).

A subset of CpGs exhibiting robust pyrosequencing assays (cg09949906, cg11993350, cg24071943) were selected for pyrosequencing validation. Primers were designed using PyroMark Design software (Qiagen) (Supplementary Data 3). A first run of pre-PCR was necessary to amplify the region including cg11993350, cg24071943 ("PCR_out", Supplementary Data 3). One microliter of BS-DNA (~10 ng) was applied as a template in the PCRs performed with the PyroMark PCR kit (Qiagen) using 5′-biotinylated reverse primers. The entire PCR product, 4 pmol of the respective sequencing primer, and streptavidin sepharose high-performance beads (GR Healthcare), were used for pyrosequencing on the PSQ 96 system and PyroMark Gold 96 reagent kit (Qiagen). The PyroMark CpG software 1.0.11 (Qiagen) served for data analysis. Assumption of equal variance between groups was tested using Brown-Forsythe test. Differences in methylation levels between DRB1*15:01 homozygotes, heterozygotes and non-carriers were tested using Kruskal-Wallis test with Dunn's multiple comparison test in GraphPad Prism 6.

A larger fragment including exon 2 of HLA-DRB1 was selected to further validate methylation differences. Knowing that this locus represents a CpG rich region and that bisulfite conversion results in a fragmentation of the DNA, amplification of the final product (~675 bp comprising 43 CpGs, 16 of them being annotated in the 450K array) required a 3-step nested-PCR protocol using SupraTherm Taq (GeneCraft, Germany) and the primers listed in Supplementary Data 3. The last PCR primers contain HindIII and EcoRI restriction enzyme motifs to allow subsequent cloning. The first 2 PCRs (PCR_A, Supplementary Data 3) were conducted according to specific cycle parameters[66] and all PCR products were diluted 1:100 prior to next PCR. After gel excision and extraction of the target band using a gel extraction kit (Qiagen), the exon 2 of HLA-DRB1 was cloned into the pcDNA3.1 vector using DH5α competent cells (One Shot® MAX Efficiency® DH5α™-T1$^R$, Life Technologies). Positive clones were verified by PCR and by restriction enzyme digestion and the plasmids were sent for sequencing (Eurofins MWG Operon, Germany). Sequence alignment was performed with Vector NTI software (InforMax).

Individual CpG allele-specific DNA methylation assessment was performed using methyl-sensitive restriction enzyme-qPCR (MSRE-qPCR). The CpG methylation-sensitive digestion of genomic DNA was carried out with the EpiJET DNA Methylation Analysis Kit based on MspI/HpaII digestion (ThermoScientific) using 30 ng of genomic DNA. Briefly, HpaII cuts only unmethylated CCGG motif from cg0603247 whereas MspI cut both unmethylated and methylated CCGG equally. The allele-specific methylation levels were quantified using the aforementioned allele-specific primers-qPCR and the 2-ΔCt method and expressed as the ratio between HpaII-digested DNA (target) and input/non-digested DNA (reference) used for enzymatic reaction. DRB1*15:01 methylation was compared to methylation of the other allele in DRB1*15:01 heterozygotes using the Mann–Whitney test in GraphPad Prism 6.

Investigation of the regulatory properties of the identified DMR was conducted using in vitro DNA methylation reporter assays. A fragment, including exon 2 of DRB1*15:01 and the regions upstream and downstream of exon 2 (1133 bp), was amplified using specific primers (Supplementary Data 3), containing AvrII or SpeI sites. PCR-amplified products were cloned into pCpG-free promoter vector (Invivogen) containing a Lucia luciferase reporter gene and into a pCpG-free basic vector (Invivogen) containing a murine secreted embryonic alkaline phosphatase (mSEAP) reporter gene. In order to address the impact of DNA methylation on the reporter gene expression, all constructs were either completely methylated (50 CpG sites) or partly methylated (methylation of internal cytosine residues in the GCGC sequence, 5 CpG sites) by incubation with either SssI or HhaI methyltransferases, respectively (New England BioLabs). The control condition (mock-methylated) was treated equally, but in absence of any methyltransferase, and corresponds to the hypomethylated DRB1*15:01 sequence. Methylated and mock-methylated constructs were purified using a QIAquick PCR purification Kit (Qiagen). The completeness of methylation was checked using an EpiJET DNA Methylation analysis Kit (MspI/HpaII) (ThermoFisher Scientific), followed by gel electrophoresis (Supplementary Fig. 1f). Original insert-empty Lucia and SEAP vectors completely, partly and mock-methylated were used as controls. For reporter gene expression assay, Human embryonic kidney HEK293T cells (kindly provided by A. Espinosa's lab, Karolinska Institutet, Sweden) were cultured as PBMCs but in Dulbecco's Modified Eagle's medium. Cell line authentication was not tested as they were used to study the effect of exogenous transfected DNA. Mycoplasma contamination was not tested. In a 96-well plate, 100 ng of constructs were co-transfected with 5-10 ng of the control vector pGL4-TK-hH Luc constitutively expressing Renilla luciferase using Effectene Transfection Reagent (Qiagen). Lucia,

SEAP and Renilla luciferase activity were measured after 48 h using QUANTI-Luc (Invivogen), the Phospha-Light System (Applied Biosystems) and the Dual-Glo Luciferase Assay System (Promega), respectively, according to manufacturer's instructions, on the GloMax 96 Microplate Luminometer (Promega). Direct or inverted orientation of the sequence yielded similar results. Results are expressed as relative activity (Lucia or SEAP normalized against Renilla) and represent the mean value of five replicates in a representative experiment replicated at least 2–3 times. Assumption of equal variance between groups was tested using Brown-Forsythe test. Statistical analyses were performed using ANOVA with Turkey's multiple comparison test in GraphPad Prism 7.

**Genome-wide DNA methylation analysis in monocytes.** We estimated the methylation level in sorted cells as the ratio between the methylated signal and the sum of methylated and unmethylated signals[67]. We filtered probes that (i) contain known common SNPs outside *HLA*, (ii) are located on the X and Y chromosomes and (iii) with detection *p*-values larger than 0.01. The sorted cell types were confirmed using the estimateCellCounts function from the minfi package[27]. To pre-process raw beta values before differential methylation analysis, we used the 3-step pipeline considered as optimal in Marabita et al.[67]: quantile normalization and Beta Mixture Quantile dilation[67]. Normalization with ComBat from the SVA package was used to correct for slide effects as identified using PCA[68]. Batch corrected beta values and Limma (Linear Models for Microarray Data)[69] were used to compute differential methylation. The linear model included age, sex, and disease stage (RRMS and SPMS) as covariates. DMR analysis was carried out using the Bumphunter package and included the same model of age, sex, and disease stage (RRMS and SPMS) as covariates. All *HLA-DRB1*-associated CpGs were subsequently tested for association with the *DRB1*15:01* genotype using additive, dominant and recessive models and correcting for sex, age, and status (HC, RRMS, or SPMS).

**Methylation mediation analysis in peripheral blood.** The methylation data from 450K arrays was preprocessed using the Illumina default procedure implemented in the Bioconductor minfi package[27]. The probe level raw data for each sample were normalized using Illumina's control probe scaling procedure and converted to methylation values on the 0–1 scale ($M/(M + U + 100)$, where $M$ and $U$ represent the methylated and unmethylated signal intensities, respectively). For DMR analysis (collapsed CpGs analysis), we averaged measurements from CpG islands, shores, and shelves into one value for each sample using the cpgCollapse function in minfi package[27]. These collapsed methylation measurements were then used for genotype-dependent DMR identification and CIT analysis, as we did previously in the single CpG analyses[19]. All analyses were performed in R 2.14 and Bioconductor 2.9.

Cell counts for the six major cell types in blood (Granulocytes, B cells, CD4$^+$ T cells, CD8$^+$ T cells, monocytes, and NK cells) for each individual were estimated using the estimateCellCounts function in minfi package[27], which obtain sample-specific estimates of cell proportions based on reference information on cell-specific methylation signatures[70].

To identify the DMRs associated with the MS phenotype, we used the bumphunter function in minfi package[27] with adjustment for confounders: age, sex, self-reported smoking status (ever smokers vs. never smokers), hybridization date, and the first two principle components of estimated differential cell counts. Regions, that have a family wise error rate (FWER) less than 0.05 with 1000 resamples and contain at least 2 probes, were identified as MS-associated DMRs. To evaluate the effect of the type of disease (RRMS, PPMS, and SPMS) or treatment at sampling (MS without treatments, MS with treatment) on DNA methylation at identified DMRs, the differences of group means were tested using either an ANOVA test (for the type of disease), or Student's *t*-test (for treatment).

To identify genotype-dependent DMRs and CIT analysis, all MS-associated DMRs were subsequently tested for association with genotype (594,262 SNPs) using an additive minor-allele dosage model. Genotype-DMR associations were corrected for multiple testing using a stringent Bonferroni-adjusted threshold of 0.05 (7 DMRs × 594,262 SNPs) = $1.20 \times 10^{-8}$. Briefly, each of the genotype (SNP)-methylation (DMR)-phenotype (MS) relationships was assessed using the CIT[71] to classify them as methylation mediated, methylation consequential, and independent. Because the CIT was designed for continuous phenotypes rather than case-control studies, we used a modified version based on logistic regression[19].

**Expression analysis from RNA-seq in PBMCs.** RNA was extracted from PBMCs from individuals that had been genotyped on the ImmunoChip and diagnosed with either MS, CIS or NINDs. Library preparation was conducted for samples that passed the criteria of sufficient amount, concentration, and quality (>of RNA. A total number of 17 samples were later discarded after quality control of sequencing data. A final number of 156 samples were included, after matching with genotype data from the same individuals (cohort 3, Supplementary Data 1). cDNA libraries for sequencing were prepared using Illumina TruSeq kit (Illumina, San Diego, USA) and sequenced on an Illumina HiSeq 2000 machine. We generated data in fastq format using Illumina 1.8 quality scores. We obtained paired-end reads with a length of 100 bp from all 156 samples on an average sequence depth of 36 million reads per sample. The reads were mapped to the H. Sapiens reference genome (NCBI v37, hg19) using STAR aligner. Conditional Quantile Normalization (CQN) method was used to normalize the count datasets and to account for the GC content bias. The residuals obtained after correcting for batch-effect and disease-

type using lmFit and eBayes from limma package[69] on CQN values is used to correlate MS risk genotypes obtained from Illumina immunochip[8] and Human660-Quad chips[4]. eQTLs were identified using a linear additive risk allele dosage model and the residual gene expression values for *HLA-DRB1*.

**Mendelian randomization.** To test the hypothesis that DNA methylation causes changes in gene expression, we exploited a two-sample MR framework. For 42 SNPs that significantly ($p \leq 5.86 \times 10^{-9}$) associated with DNA methylation at DMR3 and DMR4 and significantly ($p < 0.05$) mediated the effect of the genetic risk of MS using the CIT, we extracted the exposure summary statistics (effect size, standard error, sample size, *p*-value) from the linear regression analysis on cohort 2 (blood) using an additive minor allele dosage model. We obtained the corresponding outcome summary statistics from the *HLA-DRB1* eQTLs considered in cohort 3 (PBMC), using an additive risk allele dosage model. The MR analysis was performed using the TwoSampleMR[30] and MendelianRandomization R libraries. We firstly clumped the meQTL SNPs by removing the SNPs in strong LD ($R^2 > 0.8$ in 10,000 kb window) and then we retrieved the corresponding eQTL SNPs and associated summary statistics. We harmonized the effects of the SNP on the outcome and the exposure by ensuring that they refer to the same allele, correcting the strand for non-palindromic SNPs, and dropping all palindromic SNPs from the analysis. We then retrieved an LD correlation matrix from the MR-base database (~500 Europeans in 1000 genomes data)[30] and performed MR Egger regression accounting for the correlation between variants. This method is valid under the assumption that the associations of the genetic variants with the exposure are independent of the direct effect of the genetic variants on the outcome, and it allows us to obtain a consistent causal effect estimate as the slope from the Egger regression, as explained in Bowden et al.[29]. Noteworthy, there was no evidence of a directional pleiotropic effect ($p = 0.37$ and 0.41 for DMR3 and DMR4, respectively) or heterogeneity (Cochran's Q statistic, $p = 0.34$ and 0.57 for DMR3 and DMR4, respectively). As the method considers that all the variants must be orientated such as the associations with the exposure have the same (positive) sign, we set all the associations with the exposures to be positive and the associations with the outcome were re-oriented. The MR Steiger test for directionality was considered for assessing the correct direction of causality[30], for the variant with the strongest meQTL and eQTL effect (rs3132946). Briefly, this method performs the Steiger test to orient the direction of causality and explores a range of potential values of measurement error in the exposure and the outcome, to assess how reliable the inference of the causal direction is and gives a reliability ratio ($R$) for sensitivity analyses. The calculated $R = 6.5$ and $R = 7.9$ for DMR3 and DMR4, respectively (Supplementary Fig. 3), means that it is $R$ times more likely that the inferred direction of causality is correct (i.e., DNA methylation causes gene expression) compared to the opposite direction (gene expression causes methylation).

**Regression analyses.** We fit linear regression model with the methylation at each DMR, generated by CIT analysis in blood (cohort 2), as the outcome and the following predictors: SNP (rs9267649 or rs2227956), age, sex, self-reported smoking status (ever smokers vs. never smokers), batch, status (MS, HC), and the first two principle components of estimated differential cell counts in *DRB1*15:01* homozygotes, heterozygotes, and non-carriers separately. Analysis was performed on 279 individuals from cohort 2 and additional 333 healthy individuals from the EIRA cohort. We fit linear regression model with the *HLA-DRB1* expression value in PBMCs (cohort 3) as the outcome and the following predictors: SNP (rs9267649 or rs2227956), batch and status (MS, CIS, NINDC) in *DRB1*15:01* homozygotes, heterozygotes, and non-carriers separately. All analyses were done in Rcmd.

**Conditional association analyses and meta-analysis.** Genotyping for the different samples in the SCAND cohort was carried out using different Illumina genotyping chips and quality control was carried out for each cohort separately. The genotyping and basic quality control for the Norwegian and Danish cohort ($n$=1362) as well as 648 of the Swedish MS cases and controls has been carried out by IMSGC[4]. 12,479 of the Swedish cases and controls were genotyped using the Illumina OmniExpress bead chip by deCODE. The TwinGene cohort ($n = 6748$) was genotyped with Illumina OmniExpress bead chip at the SNP&SEQ Technology Platform Uppsala, and only one individual in each twin pair was included in subsequent analysis. Genotypes from ancestry informative markers from all the autosomes ($n = 3062$) were combined and used for identifying related individuals ($n = 1335$) using the genome command in PLINK v1.9 in a combined analysis of all the cohorts[72]. Population outliers identified using the SmartPCA program with standard settings ($n = 199$) were removed[73]. Four principal component vectors were significant in the remaining individuals and were used for controlling population stratification.

Genotypes in the *HLA* region were used to impute *HLA* genotypes using HLA*IMPv2[74]. Since genotypes for all DNA methylation-mediated CIT SNPs were not available, we replaced the missing SNPs with SNPs in high LD in the 1,000 Genomes study in the European population [http://browser.1000genomes.org/index.html]. The association of CIT markers conditional on established *HLA* MS risk alleles[5] and four principal component vectors was performed using the GLM command in R3.3.3. The established MS risk *HLA* model used contained the following terms: *DRB1*15:01* presence, *DRB1*15:01* homozygotes, *DRB1*13:03* presence, *DRB1*03:01* presence, *DRB1*03:01* homozygotes, *DRB1*08:01* presence,

*DQB1\*03:02* presence, *DQA1\*01:01* presence, *DQB1\*03:01* presence, *A\*02:01* presence, *A\*02:01* homozygotes, *B\*44:02* presence, *B\*38:01* presence, *B\*55:01* presence, two SNPs in the *HLA* region (rs2229092 and rs9277565, in both cases presence of minor allele) as well as interaction terms between presence of *DRB1\*15:01* and *DQA1\*01:01* and presence of *DQB1\*03:02* and *DQB1\*03:01*. In analyses conditioned on *DRB1\*15:01*, *DRB1\*15:01* presence, *DRB1\*15:01* homozygotes and interaction term between presence of *DRB1\*15:01* and *DQA1\*01:01* were included.

The genotyping on SWE cohort was carried out using MS replication chip, a custom-made Illumina chip with 19,000 markers in the extended *HLA* region. Related individuals ($n = 26$) and population outliers ($n = 10$) were identified as described for the SCAND cohort. Three PCA vectors were significant and used for correcting population stratification. *HLA* imputation and statistical analysis was performed in the same way as for the SCAND cohort.

The German cohort was genotyped using Illumina Omni-family microarrays. Detailed characterization and quality control of the German cohort has been published previously[9] (as cohort DE1 within the referenced publication). *HLA* alleles were imputed from genotype data using HIBAG v1.6.0[75]. Alleles with a posterior probability >0.5 were converted to hard calls. Results were validated using available *HLA* typing of 442 patients from the same cohort. SNPs were imputed with SHAPEIT2 and IMPUTE2, using the 1000 Genomes Phase 1 June 2014 release as a reference panel. Conditional regression analyses were conducted in R 3.3, using imputed *HLA* allele counts and imputed SNP dosage data. To control for population substructure, sex and the first eight multidimensional scaling components of the genetic similarity matrix were included as covariates in the regression model.

The Icelandic samples were genotyped at deCODE genetics using Illumina HumanHap610, HumanHap660, Omni-1, HumanHap300, HumanCNV30, HumanHap1M, Omni2.5, or Omni Express bead chips and additional genotypes were derived by long-range phasing and imputation[76,77]. The Icelandic *HLA* genotype calling was performed using Graphtyper[78]. Graphtyper constructs a pangenome graph for each *HLA* gene, in which each known *HLA* allele is represented with a path in the graph. Illumina short-read sequences are then aligned to the pangenome graphs and on the basis of the graph alignments, *HLA* alleles are genotyped. Subsequently, the *HLA* allele genotypes were imputed[76]. Association testing was performed using logistic regression and *p*-values were corrected for population stratification using the method of genomic control. The correction factor was estimated as 1.127 by association testing of 10,000 randomly selected sequence variants from all over the genome.

Both random and fixed effects meta-analysis between the SCAND, SWE, DE, and ICE cohorts were carried out using the meta-analysis command in PLINKv1.9. Forest plots were generated in the rmeta package.

**Data availability**. The 450K array data from monocytes and whole blood are available in the Gene Expression Omnibus (GEO) database under accession number GSE43976 and GSE106648, respectively. The RNA-seq data from PBMCs used for analyses is provided in Supplementary Data 8. The RNA-seq data could not be deposited due to inconsistencies in the current routines regarding data processing agreements, but will be made available from the corresponding author upon request and signature of data transfer agreement. The case-control cohort data used for association analysis will be made available from the corresponding author upon request and upon signature of data transfer agreement. The Twingene data can be accessed through application to the Twingene study coordinators. The raw genotype data of the German dataset cannot be shared for privacy reasons. Aggregated summary statistics are available upon request. Summary level data from deCODE may be provided by the PI upon reasonable request.

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

## Acknowledgements

This work was supported by grants from the Swedish Research Council, the Swedish Association for Persons with Neurological Disabilities, the Swedish Brain Foundation, the Swedish MS Foundation, Petrus and Augusta Hedlunds Foundation, the Swedish AFA Insurance, Knut and Alice Wallenberg Foundation, the Stockholm County Council (ALF project), and AstraZeneca (AstraZeneca-Science for Life Laboratory collaboration). A.P.F. received grant support from NIH (DP1 ES022579). L.K. was supported by fellowship from the Margaretha af Ugglas Foundation. D.G.-C. and J.T. were supported by EU FP7 306000 STATegra. Y.L. was supported in part by the National Natural Science Foundation (grant no. 31471212). We acknowledge the International Multiple Sclerosis Genetics Consortium (IMSGC) for providing SNP genotypes used in this study and BEA core facility (Karolinska Institutet) for processing 450K array data on monocytes. The computations were performed on resources provided by SNIC through Uppsala Multidisciplinary Center for Advanced Computational Science (UPPMAX). The Norwegian MS Registry and Biobank and the MS research group in Oslo are acknowledged for access to data from Norwegian MS patients. B.H. was supported by the German research foundation in the framework of the Collaborative research group TR128, the German MS competence network, and the EU project MultipleMS. This work was supported by the German Ministry for Education and Research (BMBF) as part of the "German Competence Network Multiple Sclerosis" (KKNMS) (grant nos. 01GI0916 and 01GI0917). F.Z., H.W., and R.G. were supported by the German research foundation in the framework of the Collaborative research group TR128, the German MS competence network. The KORA study was initiated and financed by the Helmholtz Zentrum München-German Research Center for Environmental Health, which is funded by the BMBF and by the State of Bavaria. Furthermore, KORA research was supported within the Munich Center of Health Sciences (MC-Health), Ludwig-Maximilians-Universität, as part of LMUinnovativ. The collection of probands in the Heinz Nixdorf RECALL Study (HNR) (PIs: K.-H. Jöckel, R. Erbel) was supported by the Heinz Nixdorf Foundation. The genotyping of HNR probands was financed through a grant of the BMBF to M. M. Nöthen. The Dortmund Health Study was supported by the German Migraine and Headache Society (DMKG) and unrestricted grants of equal share from Almirall, AstraZeneca, Berlin-Chemie, Boehringer, Boots Healthcare, GlaxoSmithKline (GSK), Janssen-Cilag, McNeil Pharma, Merck Sharp & Dohme (MSD), and Pfizer to the University of Münster. Blood collection was done through funds from the Institute of Epidemiology and Social Medicine, University of Münster (K. Berger and J. Wellmann), genotyping was supported by the BMBF (grant no. 01ER0816). SHIP is part of the Community Medicine Research Network of the University Medicine Greifswald, Germany (www.community-medicine.de), which was initiated and funded by the BMBF (grants no. 01ZZ9603, 01ZZ0103, and 01ZZ0403), the Ministry of Cultural Affairs and the Social Ministry of the Federal State of Mecklenburg-West Pomerania; genome-wide data have been supported by the BMBF (grant no. 03ZIK012). The FoCus study was supported by the BMBF (grant no. 0315540A).

## Author contributions

M.J. conceived and designed the study. M.J., T.J.E., and A.P.F. designed and oversaw the initial mediation experiments. M.J. and I.K. supervised the analyses. L.K., Y.L., S.R., G.Z., S.A., B.G., and M.L. conducted the experiments. C.G., H.W., F.Z., R.G., B.T., F.W., B.H., KKNMS, K.S., S.H.-H., R.R., U.S., C.O.S., T.K., A.F., M.L., E.G.C., H.B.S., H.F.H., A.B.O., H.H., E.O., I.J., K.S., T.O., and F.P. collected and managed the samples and clinical information. Y.L., D.G.-C., T.J., F.M., E.E., P.S., J.L., A.T.D., T.F.M.A., S.O., H.P.E., B.V. H., I.K., and M.J. performed the analyses. L.K., A.P.F., and M.J. designed the follow-up experiments and wrote the manuscript with assistance from all authors. All authors read and approved the manuscript.

## Additional information

**Competing interests:** Sigurgeir Olafsson, Hannes P. Eggertsson, Bjarni V. Halldorsson, Ingileif Jonsdottir and Kari Stefanssonare employees of deCODE genetics/Amgen Inc at the time work related to this study was carried out.H.F. Harbo has received honoraria for advice and lecturing from Biogen, Genzyme, Merck, Novartis,Sanofi-Aventis and Teva. She has received modest unrestricted research grant for research fromNovartis. B. Tackenberg received personal speaker honoraria and consultancy fees as a speaker andadvisor from Bayer Healthcare, Biogen, CSL Behring, GRIFOLS, Merck Serono, Novartis, Octapharma,Roche, Sanofi Genzyme, TEVA und UCB Pharma. His University received unrestricted research grantsfrom Biogen-idec, Novartis, TEVA, Bayer Healthcare, CSL-Behring, GRIFOLS, Octapharma, SanofiGenzyme and UCB Pharma. B. Hemmer has served on scientific advisory boards for F. Hoffmann-LaRoche Ltd, Novartis, Bayer AG, and Genentech; he has served as DMSC member for AllergyCare and TGtherapeutics; he or his institution have received speaker honoraria from Biogen Idec, TevaNeuroscience, Merck Serono, Medimmune, Novartis, Desitin, and F. Hoffmann-La Roche Ltd; hisinstitution has received research support from Chugai Pharmaceuticals and Biogen; holds part of two patents; one for the detection of antibodies against KIR4.1 in a subpopulation of MS patients and one for genetic determinants of neutralizing antibodies to interferon β. The remaining authors declare no competing interests.

Lara Kular[1], Yun Liu[2,3], Sabrina Ruhrmann[1], Galina Zheleznyakova[1], Francesco Marabita[1], David Gomez-Cabrero[4], Tojo James[1], Ewoud Ewing[1], Magdalena Lindén[1], Bartosz Górnikiewicz[1], Shahin Aeinehband[1], Pernilla Stridh[1], Jenny Link[1], Till F.M. Andlauer[5,6,7], Christiane Gasperi[6,7], Heinz Wiendl[7,8], Frauke Zipp[7,9], Ralf Gold[7,10], Björn Tackenberg[7,11], Frank Weber[5,7], Bernhard Hemmer[6,7,12], Konstantin Strauch[13], Stefanie Heilmann-Heimbach[14], Rajesh Rawal[15], Ulf Schminke[16], Carsten O. Schmidt[17], Tim Kacprowski[18], Andre Franke[19], Matthias Laudes[20], Alexander T. Dilthey[21,22], Elisabeth G. Celius[23,24], Helle B. Søndergaard[25], Jesper Tegnér[4,26], Hanne F. Harbo[23,27], Annette B. Oturai[25], Sigurgeir Olafsson[28], Hannes P. Eggertsson[28], Bjarni V. Halldorsson[28,29], Haukur Hjaltason[30,31], Elias Olafsson[30,31], Ingileif Jonsdottir[28,31,32], Kari Stefansson[28,31], Tomas Olsson[1], Fredrik Piehl[1], Tomas J. Ekström[1], Ingrid Kockum[1], Andrew P. Feinberg[2] & Maja Jagodic[1]

[1]Department of Clinical Neuroscience, Center for Molecular Medicine, Karolinska Institutet, 171 77 Stockholm, Sweden. [2]Center for Epigenetics, and Departments of Medicine, Biomedical Engineering and Mental Health, Johns Hopkins University, Baltimore, MD 21205, USA. [3]Key Laboratory of Metabolism and Molecular Medicine, Ministry of Education; Department of Biochemistry and Molecular Biology, Fudan University Shanghai Medical College, 200032 Shanghai, China. [4]Unit of Computational Medicine, Department of Medicine, Solna, Center for Molecular Medicine, Karolinska Institutet, 171 77 Stockholm, Sweden. [5]Max Planck Institute of Psychiatry, 80804 Munich, Germany. [6]Department of Neurology, Klinikum rechts der Isar, School of Medicine, Technische Universität München, 81675 Munich, Germany. [7]German Competence Network Multiple Sclerosis (KKNMS), Klinikum Rechts der Isar, Technische Universität München, 81675 Munich, Germany. [8]Department of Neurology with Institute of Translational Neurology, University of Münster, 48149 Münster, Germany. [9]Department of Neurology, University Medicine Mainz, Johannes Gutenberg University Mainz, 55122 Mainz, Germany. [10]Department of Neurology, St. Josef-Hospital, Ruhr-University Bochum, 44801 Bochum, Germany. [11]Neuroimmunology Center, Marburg University, 35037 Marburg, Germany. [12]Munich Cluster for Systems Neurology (SyNergy), 81377 Munich, Germany. [13]Institute of Genetic Epidemiology, Helmholtz Zentrum München, Neuherberg, Germany and Institute of Medical Informatics, Biometry, and Epidemiology, Chair of Genetic Epidemiology, Ludwig-Maximilians-Universität, 80539 Munich, Germany. [14]Institute of Human Genetics, University Hospital Bonn and Division of Genomics, Life & Brain Research Centre, University of Bonn School of Medicine, 53113 Bonn, Germany. [15]Institute of Epidemiology and Social Medicine, University of Münster, 48149 Münster, Germany. [16]Department of Neurology, University Medicine Greifswald, 17489 Greifswald, Germany. [17]Institute for Community Medicine, University Medicine Greifswald, 17489 Greifswald, Germany. [18]Interfaculty Institute for Genetics and Functional Genomics, Ernst Moritz Arndt University and University Medicine Greifswald, 17489 Greifswald, Germany. [19]Institute of Clinical Molecular Biology, Kiel University, 24105 Kiel, Germany. [20]Department I of Internal Medicine, Kiel University, 24105 Kiel, Germany. [21]Wellcome Trust Centre for Human Genetics, University of Oxford, Oxford OX3 7BN, UK. [22]Institute of Medical Microbiology and Hospital Hygiene Heinrich-Heine-University Düsseldorf, 40225 Düsseldorf, Germany. [23]Department of Neurology, Oslo University Hospital, 0372 Oslo, Norway. [24]Institute of Health and Society, Faculty of Medicine, University of Oslo, 0450 Olso, Norway. [25]Danish Multiple Sclerosis Center, Department of Neurology, Rigshospitalet, University of Copenhagen, 2100 Copenhagen, Denmark. [26]Biological and Environmental Sciences and Engineering Division, Computer, Electrical and Mathematical Sciences and Engineering Division, King Abdullah University of Science and Technology, Thuwal 23955, Saudi Arabia. [27]Institute of Clinical Medicine, University of Oslo, 0450 Oslo, Norway. [28]deCODE genetics/Amgen Inc, 101 Reykjavik, Iceland. [29]School of Science and Engineering, Reykjavik University, 101 Reykjavik, Iceland. [30]Department of Neurology, Landspitali, The National University of Iceland, 101 Reykjavik, Iceland. [31]Faculty of Medicine, School of Health Sciences, University of Iceland, 101 Reykjavik, Iceland. [32]Department of Immunology, Landspitali, The National University Hospital of Iceland, 101 Reykjavik, Iceland. These authors contributed equally: Lara Kular, Yun Liu, Ingrid Kockum, Andrew P. Feinberg, Maja Jagodic.

