## [Peer Review File · Nature Communications]

Reviewers' comments:

Reviewer #1 (Remarks to the Author):

Very interesting paper outlining the importance of methylation in the pathomechanism of MS. Methylation changes in the HLA_DRB15 region has been previously reported in CD4+ve cells of MS patients but not in monocytes and not in such a large cohort. The study was well designed and the results support the conclusion. There are, though, some questions that need to be addressed.

- The results were confirmed in a large cohort of MS patients. What happens in HLA-DR15 positive HC?
- Genome wide arrays were only done in cohort 1 and part of cohort 2
 - o It is not clear from Figure 1 how the 60 patients included were analysed by the different methods and on what criteria some were excluded.
 - o Only 450 k microarray was used, not the 850 k currently available
 - o Do we know if these patients were early or late disease?
 - o Do we know what treatment these patients were on at the time of sample collection?
- Cell separation was only done on cohort 1 (23 patients)
 - o Results are only reported on Monocytes. Did they find the same results in CD4, CD8, CD19 and NK cells? CD8 cells for example don't appear to show the same effect on HLA-DRB15 (Maltby et al., Clin Epigen 2015)
 - o Are the results from cohort 2 which used blood instead of cell subtypes the same? I don't think we can say with confidence that the signal comes only from monocytes. A deconvolution approach would be of interest to determine which cell types contribute what to the signal.

Reviewer #2 (Remarks to the Author):

This is an interesting and potentially important study if the following additions could be made:

- (1) Using antibodies demonstrate a difference in protein expression of DRB1*1501 versus X.
- (2) There is much controversy surrounding the primary role of methylation in disease susceptibility – see, for example, the recent review and criticism by Lappalainen and Greally Nat Rev Genet July 2017. How can the authors rule out the possibility that the methylation of exon 2 is a downstream event of transcription and differential binding of transcription factors (TFs). Their in vitro analyses of exon 2 are difficult to interpret. Perhaps it's a feature of transcriptional regulation of this allele and this pattern of transcription results in altered methylation? Is the methylation really a mediator, or a symptom of more primary events?
Can the authors please discuss.
- (3) How do their results overlap, or not, with other recently published methylation studies: Bell CG et al Nat Com 2018; Bonder MJ et al Nat Genet January 2017?
- (4) After conditioning for DRB1*1501, they report in MS a protective association. This is interesting and would be even more impactful if they sought evidence of association in other autoimmune diseases that have associations with DRB1*1501, such as type 1 diabetes in which one might predict that this MS variant is predisposing to T1D (DRB1*1501 is protective in T1D)?

Point-by-point reply to reviewers' comments (NCOMMS-18-00865A)

Reviewer #1 (Remarks to the Author):

Very interesting paper outlining the importance of methylation in the pathomechanism of MS. Methylation changes in the HLA_DRB15 region has been previously reported in CD4+ve cells of MS patients but not in monocytes and not in such a large cohort. The study was well designed and the results support the conclusion. There are, though, some questions that need to be addressed.

Response: We thank the reviewer for a positive and thorough review of our manuscript that has improved our findings and clarified our results and interpretations.

Comment #1: The results were confirmed in a large cohort of MS patients. What happens in HLA-DR15 positive HC?

Response: The methylation status of the differentially methylated regions (DMRs) in the *HLA* locus is genotype-driven, i.e., the *DRB1*15:01* allele status largely determines the methylation levels, irrespective of the case-control status. This can be observed in cohort 2 (blood, n=140 MS and n=139 HC) where *DRB1*15:01* exerts similar effect on all seven DMRs in MS patients and HC (please see Supplementary Figure 5). We have now clarified this point in the discussion of the revised manuscript by adding the following text:

Discussion (page 12): "The impact of the *DRB1*15:01* genotype on *HLA-DRB1* methylation was similar in MS patients and healthy controls (Supplementary Fig. 5). This strong genetic effect can explain the observed methylation differences between MS patients and controls in all studied cell types, i.e., CD14⁺ monocytes, CD19⁺ B cells, CD4⁺ T cells, and CD8⁺ T cells."

Comment #2: Genome wide arrays were only done in cohort 1 and part of cohort 2. It is not clear from Figure 1 how the 60 patients included were analysed by the different methods and on what criteria some were excluded.

Response: We thank the reviewer for highlighting the lack of clarity in this section. We have up-dated the status of all individuals used in the study and clarified the selection criteria (one patient with clinically isolated syndrome and one with inflammatory component have been diagnosed with MS in the meantime). Monocytes were sorted from 82 individuals (62 MS patients and 20 healthy controls, HC) as indicated in the Supplementary Table 1. In total, 36 individuals (23 MS patients and 13 HC) were selected for the 450K array analysis based on (i) the available amount of DNA required for 450K arrays and (ii) to match, as much as possible, cases and controls regarding covariates such as age and sex. For locus-specific methylation, all samples with sufficient quantity of DNA (500 ng) were used for validation

with pyrosequencing (34 MS and 15 HC). All samples with RNA of sufficient quality were used for expression analysis with qPCR (39 MS and 19 HC). For allele-specific analyses (methylation and expression), all samples that were heterozygotes for *DRB1*15:01* and had a sufficient quantity and quality of DNA/RNA were used (also a few individuals that were homozygotes and non-carriers were included as controls, as presented in Supplementary Figure 1). Details of individuals used in specific analyses are provided in the Supplementary Table 1. We have now also included a description of the selection criteria in the revised manuscript, as follows:

Methods, Materials and sample preparation (page 15): “Cohort 1 consisted of monocytes isolated from 62 MS patients and 20 healthy controls and was used for genome-wide and locus-specific DNA methylation and expression analyses. Thirty-six of them were selected for genome-wide analysis based on the sufficient amount of DNA required for 450K arrays and the matching clinical parameters (sex and age) between the groups. Of the 23 MS patients, 91% (21/23) were not treated at the time of sampling (either never treated or they stopped treatment at least 6 months before). All samples with DNA of sufficient quantity (n=49) were used for pyrosequencing validation. All samples with available RNA of sufficient quality (n=58) were used for qPCR-based gene expression analysis. All samples that were *DRB1*15:01* heterozygotes, and where DNA/RNA of a sufficient quantity and quality was available, were used for allele-specific methylation and expression analyses. Details of individuals used in specific analyses are provided in the Supplementary Table 1.”

Figure 1: the figure has been revised to clarify the number of individuals used in the different analyses and to match the additional description in the Methods.

Comment #3: Only 450 k microarray was used, not the 850 k currently available.

Response: Only 450K arrays were used as they were the only available option at the time when these samples were processed.

Comment #4: Do we know if these patients were early or late disease?

Response: We thank the reviewer for highlighting the lack of this information that has been included in the revised manuscript.

- In cohort 1 (monocytes), a mixture of mostly recently diagnosed relapsing-remitting (RRMS) patients but with variable suspected disease duration and patients with longer disease duration, in earlier phases of a secondary-progressive stage (SPMS) was included in the 450K analysis (10 RRMS, 13 SPMS and 13 HC, as indicated in the Supplementary Table 1). The disease stage did not have any impact on the *HLA-DRB1* methylation. To account for a potential genome-wide impact, the disease stage has

been used as a covariate in the analyses, as described in the Methods in the section “Genome-wide DNA methylation analysis in monocytes” (page 19).

- In cohort 2 (blood), the majority of patients were RRMS (n=121), while only few were diagnosed with primary progressive MS (PPMS, n=4), or SPMS with long disease duration (n=15). We have now tested if the type of disease affects methylation at any of the seven DMRs in the *HLA* locus discovered by the Causal Inference Testing, and we found no evidence of any influence. This has been included in the revised manuscript as follows:

Results (page 9): “The type of disease or treatment status at sampling had no effect on the methylation levels at any of the seven DMRs ($p>0.3$ for all comparisons).”

Methods, Materials and sample preparation (page 15): “Cohort 2 used for CIT analysis on peripheral blood cells included 140 MS patients with relapsing-remitting (RRMS, n=121), primary progressive (PPMS, n=4), or secondary progressive (SPMS, n=15) disease, and 139 healthy controls.”

Methods, Methylation mediation analysis in peripheral blood (page 20): “To evaluate the effect of the type of disease (RRMS, PPMS, and SPMS) or treatment at sampling (MS without treatments, MS with treatment) on DNA methylation at identified DMRs, the differences of group means were tested using either an ANOVA test (for the type of disease), or Student's t-test (for treatment).”

Comment #5: Do we know what treatment these patients were on at the time of sample collection?

Response: We have now extracted the treatment information for the MS patients used in genome-wide DNA methylation studies and performed additional analysis.

- In cohort 1 (monocytes): 91% of the patients (21/23) used for the 450K analysis were not treated at the time of sampling (either never treated or they stopped treatment at least 6 months before). This description has been added in the revised manuscript:

Methods, Materials and sample preparation (page 15): “Of the 23 MS patients, 91% (21/23) were not treated at the time of sampling (either never treated or they stopped treatment at least 6 months before).”

- In cohort 2 (blood): 35% of patients (49/140) were not treated at the time of sampling while 65% (91/140) received some type of a disease-modifying treatment. As most of the samples were collected prior to wider adoption of more potent immunomodulatory drugs in MS (sampling between 2005-2008), the majority of treated patients 85% (77/91) received drugs that have a moderate impact on disease activity (e.g., 63 were treated with interferon beta preparations and 12 with glatiramer acetate). We have now performed analysis to test if the treatment at the

time of sampling affects the methylation at any of the seven DMRs in the *HLA* locus discovered by the Causal Inference Testing and we found no evidence of any influence. This has been included in the revised manuscript as follows:

Results (page 9): "The type of disease or treatment status at sampling had no effect on the methylation levels at any of the seven DMRs ($p > 0.3$ for all comparisons)."

Methods, Materials and sample preparation (page 15): "In total, 65% (91/140) of MS patients were treated at the time of sampling but the majority, 85% (77/91), received drugs that have a moderate impact on disease activity (e.g., 63 were treated with interferon beta preparations and 12 with glatiramer acetate)."

Methods, Methylation mediation analysis in peripheral blood (page 20): "To evaluate the effect of the type of disease (RRMS, PPMS, and SPMS) or treatment at sampling (MS without treatments, MS with treatment) on DNA methylation at identified DMRs, the differences of group means were tested using either an ANOVA test (for the type of disease), or Student's t-test (for treatment)."

Comment #6: Cell separation was only done on cohort 1 (23 patients). Results are only reported on Monocytes. Did they find the same results in CD4, CD8, CD19 and NK cells? CD8 cells for example don't appear to show the same effect on HLA-DRB15 (Maltby et al., Clin Epigen 2015).

Response: Indeed, after discovering the seven DMRs in blood, we investigated several specific cell types that are implicated in the pathogenesis of MS. In addition to CD14⁺ monocytes, we investigated CD4⁺ and CD8⁺ T cells as well as CD19⁺ B cells sorted from blood of MS patients and healthy controls (details of the individuals included in this analysis are given in the Supplementary Table 1). We have not studied less frequent cell types, such as NK cells, as no sufficient DNA yield of these cell types for 450K analysis could be obtained from our sorting set-up. We observed significant differences in six of the seven DMRs in all four sorted cell types (please see Figure 4 and Supplementary Table 5). This is in line with the strong effect of the *DRB1*15:01* genotype on *HLA-DRB1* methylation, as demonstrated by us and others in multiple tissues, such as white blood cells (Hong *et al.*, Nat Commun. 2015), pancreatic islets (Olsson *et al.*, PLoS Genet. 2014), and brain samples (Do *et al.*, Am J Hum Genet. 2016). Since the *DRB1*15:01* frequency is much higher in MS patients than controls, the genotype-driven methylation changes can explain the observed differences between cases and controls in multiple cell types.

We could not find a sufficiently detailed description (e.g., the final list of analyzed CpGs and information regarding potential exclusion criteria of individuals) to directly compare our findings to those of Maltby *et al.*. In addition, none of the 79 significant CpGs detected by Maltby *et al.* overlap with the top 40 candidate CpGs detected by Bos *et al.* in CD8⁺ T cells (Bos *et al.*, PLoS One. 2015). While Bos *et al.* have filtered out the majority of *HLA* probes (as

they have not focused on this locus that requires additional technical validation, as extensively done here), the two remaining CpG probes that overlap with our DMR2 and DMR5 show the same direction of effect between MS and HC as in our study, and are nominally significant both in CD4⁺ and CD8⁺ cells. In the revised manuscript, we discuss several other studies that have identified significant genotype-driven (focusing on *DRB1*15:01*) methylation changes in the *HLA* locus in multiple cell types and tissues (for details please see our response to **Comment #3 from Reviewer #2**). Since we did, however, observe the least pronounced differences in CD8⁺ T cells, this has been clarified in the revised text:

Results (page 9): “We further replicated the methylation differences between MS cases and controls for six out of the seven DMRs using sorted CD14⁺ monocytes (n=36, cohort 1), CD19⁺ B cells (n=29), as well as CD4⁺ (n=33) and CD8⁺ (n=29) T cells, with the least pronounced differences being observed in CD8⁺ T cells (Fig. 4c, Supplementary Table 5).”

Comment #7: Are the results from cohort 2 which used blood instead of cell subtypes the same? I don't think we can say with confidence that the signal comes only from monocytes. A deconvolution approach would be of interest to determine which cell types contribute what to the signal.

Response: In blood cohort 2, as correctly pointed out by the reviewer, we do not know from what cell type the signal comes from. However, in order to remove the potential influence of differences in cell type composition on methylation analyses, we estimated the six major cell type proportions in each sample based on cell type-specific methylation signatures and adjusted accordingly in the regression analyses (this has been described in detail in the Methods, page 20). Even though this well-known deconvolution approach can address some confounding effects from cell types (Liu *et al.*, Nat Biotechnol. 2013), it does not address the respective contribution from each cell type to the signal (this is, in fact, an active research area and, to our knowledge, there are currently no statistical models available for estimating the respective contributions of individual cell types present in blood). However, our findings strongly suggest that the methylation differences which we observed in blood are mainly driven by the genotype and that these differences do not only originate from monocytes. In fact, the methylation differences were replicated in all four sorted cell types (please see Figure 4C, top right, and Supplementary Table 5), as also stated in the manuscript (page 9). For better clarity, we have added the following text in the discussion of the revised manuscript:

Discussion (page 12-13): “The impact of the *DRB1*15:01* genotype on *HLA-DRB1* methylation was similar in MS patients and healthy controls (Supplementary Fig. 5). This strong genetic effect can explain the observed methylation differences between MS patients and controls in all studied cell types, i.e., CD14⁺ monocytes, CD19⁺ B cells, CD4⁺ T cells, and CD8⁺ T cells. Similar to our findings, *HLA-DRB1* methylation differences between MS patients

and controls have been reported in CD4⁺ T cells and suggested to be partially dependent on *DRB1*15:01*¹⁴. Previous meQTLs studies have reported strong genetic regulation of methylation at CpGs shared by the DMRs identified in our study in white blood cells²⁴, pancreatic islets²³, and brain samples⁴⁴. Since these studies have not focused on *DRB1*15:01*, we have investigated the SNPs underlying the reported meQTLs and we found that many are in high LD with *DRB1*15:01* and show the same direction of effect as in our study. It is thus not surprising that *HLA-DRB1* methylation differences between cases and controls exist in multiple cells types, although the functional consequences of this genotype-driven methylation likely differ between distinct cell types and depend on the contribution of HLA class II molecules to a particular cell type-specific function.”

Reviewer #2 (Remarks to the Author):

This is an interesting and potentially important study if the following additions could be made:

Response: We thank the reviewer for a positive and constructive review of our manuscript that has improved interpretations of our findings.

Comment #1: Using antibodies demonstrate a difference in protein expression of *DRB1*1501* versus X.

Response: We thank the reviewer for this comment, pointing at the translation from mRNA to protein levels. However, the interpretation of HLA class II protein expression is particularly challenging compared to other classes of molecules and, so far, it is not feasible to address this question in the relevant context. Technical difficulties to investigate the specific HLA molecules exist (point 1) and the complexity of the antigen presentation process is a further complication (point 2):

(1) Allele-specific investigation of the HLA-DRB1 protein surface expression is technically challenging due to the lack of validated commercially available antibodies that are specific for *DRB1*15:01*-encoded proteins. Available antibodies rather target HLA-DR complexes, i.e., all HLA-DRA/DRB complexes, and not specifically HLA-DRA/DRB1. Analyses are therefore biased by the presence of other HLA-DR complexes in a haplotype-specific manner. For example, in *DRB1*15:01* carriers, the staining would reflect the surface expression of HLA-DRB5 in addition to HLA-DRB1 molecules (as they are present on the same haplotype). In line with this, cases of antibody cross-reactivity against several HLA-DRB complexes have been observed (Marrari and Duquesnoy, 2009), further highlighting the need for development and experimental validation of specific antibodies.

(2) The translation from *HLA-DRB1* transcriptional expression to presence of receptor complexes at the cell surface is not straightforward, since translocation to the surface involves peptide binding to HLA class II antigen presenting groove. Knowing that new HLA molecules are constantly being synthesized, and remain ready to accept peptides, our data indicate that higher *DRB1*15:01* transcript levels increase the probability of them binding to the peptide and to present it in higher amounts on the surface. In the context of MS pathogenesis, this is most likely the case for specific antigen presenting cells (APCs) presenting MS autoantigen peptide(s) and may not be detected using conventional FACS strategies.

Therefore, the interpretation of the HLA class II protein surface expression from transcriptional data is not trivial, given the lack of validated experimental tools, the complexity of antigen presentation, the immunology of MS, as well as the unknown autoantigen and APC population.

Comment #2: There is much controversy surrounding the primary role of methylation in disease susceptibility – see, for example, the recent review and criticism by Lappalainen and Greally Nat Rev Genet July 2017. How can the authors rule out the possibility that the methylation of exon 2 is a downstream event of transcription and differential binding of transcription factors (TFs). Their *in vitro* analyses of exon 2 are difficult to interpret. Perhaps it's a feature of transcriptional regulation of this allele and this pattern of transcription results in altered methylation? Is the methylation really a mediator, or a symptom of more primary events? Can the authors please discuss.

Response: The reviewer is raising indeed a highly relevant issue of a possible reverse causation, which would, in this case, refer to DNA methylation changes induced by transcriptional activity. In accordance with suggestions from Lappalainen and Greally towards “Second-generation EWAS”, we have tried to address this issue both experimentally and analytically in the original manuscript:

(1) We first used two experimental approaches, aiming at manipulating DNA methylation in the *HLA-DRB1* gene and evaluating the subsequent transcriptional outcome. *Ex-vivo*, we used the well-known demethylating drug 5-aza-deoxycytidine on cultured PBMCs from *DRB1*15:01* non-carriers (i.e., having a high *HLA-DRB1* methylation) and observed an increased *HLA-DRB1* gene expression after treatment (depicted in Fig. 3C). We then utilized an *in-vitro* reporter system, where we induced different degrees of methylation at the sequence encompassing exon 2 of the *DRB1*15:01* allele and monitored consequential expression. This experimental strategy allowed us to show (i) the regulatory feature of the sequence encompassing exon 2 of the *HLA-DRB1* gene on transcription and (ii) a direct effect of inducing DNA methylation at the region encompassing exon 2 on reduced transcriptional activity. The directionality of the

impact is, in the case of the reporter system, experimentally validated. To clarify this, we have now modified the corresponding text with the following additions:

Results (page 8): “We tested whether methylation in the region encompassing exon 2 of *DRB1*15:01* can exert regulatory properties on gene expression. We addressed enhancer and promoter activity and the effect of DNA methylation levels of the region using methylation-sensitive CpG-free vector-based reporter systems. The inserts were methylated using two different methyltransferases, *SssI* and *HhaI*, which methylate all CpG sites (50 CpGs) or only the internal cytosine residue in GCGC (5 CpGs), respectively. Notably, the region encompassing exon 2 of *DRB1*15:01* displayed significantly decreased enhancer ($p=1.3 \times 10^{-4}$) and promoter ($p=9.7 \times 10^{-3}$) activity if the insert was fully methylated (Fig. 3f-g, Supplementary Fig. 1f). This indicates that DNA methylation is a regulatory feature of the region encompassing exon 2 of *DRB1*15:01* and that the hypomethylated *DRB1*15:01* has a capacity to drive higher gene expression.”

Methods, In-vitro DNA methylation reporter assays (page 18): “In order to address the impact of DNA methylation on the reporter gene expression, all constructs were either completely methylated (50 CpG sites) or partly methylated (methylation of internal cytosine residues in the GCGC sequence, 5 CpG sites) by incubation with either *SssI* or *HhaI* methyltransferases, respectively (New England BioLabs). The control condition (mock-methylated) was treated equally, but in absence of any methyltransferase, and corresponds to the hypomethylated *DRB1*15:01* sequence.”

(2) Second, we addressed reverse causation analytically, using a method combining a two-sample Mendelian Randomization (MR) framework (with Egger regression for causal inference) and a Steiger test for inference of directionality. Indeed, in line with the criticism from Lappalainen and Greally, it has been recently shown that the reliability of causal inference can be questioned in the presence of measurement error, which is plausible in *omics* measurement of DNA methylation and gene expression. In that context, MR-Steiger has been recently introduced to use summary-level statistics and test causal relationships, by triangulating the problem with MR and a test for directionality (Steiger Test) (Bowden *et al.*, Int J Epidemiol. 2015; Hemani *et al.*, PLoS Genet. 2017). This strategy has been shown to compare favorably to other methods in assessing the direction of causality, especially in presence of measurement error (Hemani *et al.*, PLoS Genet. 2017). Therefore, we combined MR with a Steiger test (MR-Steiger) to assess the direction of causality, and we present a sensitivity analysis for the variant with the strongest meQTL and eQTL effect (rs3132946, Supplementary Fig. 3). The calculated reliability values (R) presented in Supplementary Figure 3, i.e., $R=6.5$ and $R=7.9$ for DMR3 and DMR4, respectively, mean that it is R times more likely that the inferred direction of causality is correct (i.e., DNA methylation causes gene expression) than the opposite direction (gene expression causes methylation). The other scenario under which the Steiger test would support the incorrect causal direction is the presence of a pleiotropic effect of the instrumental variables (meQTL/eQTL). However, we have

shown (see Methods) that there is no evidence of a directional pleiotropic effect. Nevertheless, we apologize if our approach was not sufficiently described and we have modified now the corresponding Method section and the legend of Supplementary Figure 3.

Discussion (page 13): “Since DNA methylation changes can actively impact gene expression or be a consequence of transcriptional activity in the locus⁴⁵, we further addressed the potential causal effect of DNA methylation in exon 2 of *HLA-DRB1* on expression both experimentally and analytically. Using an *in-vitro* reporter system, we demonstrated that the exon 2 sequence of *DRB1*15:01* exerts regulatory properties on gene expression in a DNA methylation-dependent manner. This suggests that hypomethylation of exon 2 could mediate the effect of *DRB1*15:01* on *HLA-DRB1* gene expression. This is also supported by the significant causal relationship between methylation at DMRs in exon 2 and *HLA-DRB1* expression in PBMCs, obtained using two-sample MR. The directionality of the causal relationship was further confirmed using a method that combines MR with a Steiger test (MR-Steiger)³⁵.”

Methods, Mendelian Randomization (MR) (page 21-22): “The MR Steiger test for directionality was considered for assessing the correct direction of causality³⁵, for the variant with the strongest meQTL and eQTL effect (*rs3132946*). Briefly, this method performs the Steiger test to orient the direction of causality and explores a range of potential values of measurement error in the exposure and the outcome, to assess how reliable the inference of the causal direction is and gives a reliability ratio (R) for sensitivity analyses. The calculated $R=6.5$ and $R=7.9$ for DMR3 and DMR4, respectively (Supplementary Fig. 3), means that it is R times more likely that the inferred direction of causality is correct (i.e., DNA methylation causes gene expression) compared to the opposite direction (gene expression causes methylation).”

Legend of Supplementary Figure 3, MR Steiger and sensitivity analyses: “MR Steiger and sensitivity analyses for DMR3 (a) and DMR4 (b). The x and y axis represent different potential values of measurement error for the exposure (methylation) and the outcome (expression), respectively. The z axis shows the value of the MR Steiger test under measurement error (blue surface). When the correlation between the outcome and the variant (ρ_{gy}) is lower than the correlation between the exposure and the variant (ρ_{gx}), the values on the z axis are negative and the MR-Steiger test would infer the correct direction of the causality. Therefore, the MR-Steiger test supports the inferred direction of causality when the blue surface is below the black plane. Conversely, the test supports the wrong direction of causality when the blue surface is above the black plane. We can consider the positive and negative volumes bound by the blue and the black surface and calculate a reliability value (R) as the ratio between the negative and the positive volumes. The calculated $R=6.5$ and $R=7.9$ for DMR3 and DMR4, respectively, mean that it is R times more likely that the inferred direction of causality is correct (i.e., DNA methylation causes gene expression), compared to the opposite direction (gene expression causes methylation). ”

Comment #3: How do their results overlap, or not, with other recently published methylation studies: Bell CG et al Nat Com 2018; Bonder MJ et al Nat Genet January 2017?

Response: We thank the reviewer for the suggestion to compare our findings with other studies that have looked at genotype-driven methylation changes. It is however challenging to compare findings affecting the *HLA* locus to studies that have not focused on *HLA* and thus do not take the complexity of the locus into consideration. More specifically:

- In genome-wide approaches using 450K arrays (such as in Bonder *et al.*, Nat Genet. 2017), the probes overlapping with the positions of known common SNPs located within the probe, or which are in high LD with a SNP inside the probe, are typically removed during data processing. With this generic approach, too few coarse probes remain in the *HLA* locus to permit identification of differentially methylated regions.
- In genome-wide approaches using sequencing methodologies including MeDIP-seq (such as in Bell *et al.*, Nat Commun. 2018) and whole genome bisulfite sequencing (Guo *et al.*, Nat Genet. 2017), all reads with low mappability are typically excluded before analyses. This may cause the lack of association to DNA methylation at the *HLA* class II locus in these studies.

However, despite the technical limitations of addressing DNA methylation in *HLA* class II genes, other genome-wide studies have reported meQTLs including CpGs that overlap with our DMRs. In the studies showing meQTLs in the *HLA* class II locus and for which the direction of the effect is reported, we found overall similar impact of SNPs in LD with *DRB1*15:01* (sometimes the same SNPs as in our study are reported) on methylation in different tissues, i.e., peripheral white blood cells (Hong *et al.*, Nat Commun. 2015), pancreatic islets (Olsson *et al.*, PLoS Genet. 2014), and brain tissue (Do *et al.*, Am J Hum Genet. 2016). For instance, out of the 50 SNPs affecting MS risk through DNA methylation at *HLA* class II genes reported in our study, 11 were shown to similarly associate with decreased methylation at CpGs in DMR3 and DMR4 in pancreatic islets (Olsson *et al.*, PLoS Genet. 2014). We have now detailed this point in the discussion of the revised manuscript:

Discussion (page 13): “Previous meQTLs studies have reported strong genetic regulation of methylation at CpGs shared by the DMRs identified in our study in white blood cells²⁴, pancreatic islets²³, and brain samples⁴⁴. Since these studies have not focused on *DRB1*15:01*, we have investigated the SNPs underlying the reported meQTLs and we found that many are in high LD with *DRB1*15:01* and show the same direction of effect as in our study. It is thus not surprising that *HLA-DRB1* methylation differences between cases and controls exist in multiple cells types, although the functional consequences of this genotype-driven methylation likely differ between distinct cell types and depend on the contribution of *HLA* class II molecules to a particular cell type-specific function.”

Comment #4: After conditioning for DRB1*1501, they report in MS a protective association. This is interesting and would be even more impactful if they sought evidence of association in other autoimmune diseases that have associations with DRB1*1501, such as type 1 diabetes in which one might predict that this MS variant is predisposing to T1D (DRB1*1501 is protective in T1D)?

Response: This is indeed an extremely interesting hypothesis. However, we feel that it is outside of the scope of this manuscript. We do not have access to large case-control cohorts of Type 1 Diabetes (T1D) that would be required for such conditional analyses, considering that an *HLA* association with T1D is likely mediated by multiple independent and interacting variants (Hu *et al.*, Nat Genet. 2015, Zhao *et al.*, Diabetes. 2016). Even larger cohorts would be required if there is an interaction with *DRB1*15:01*, as this haplotype is very rare among T1D patients, e.g., the frequency of *DRB1*15:01* is around 1.5% in the Swedish population (Graham *et al.*, Eur J Immunogenet. 1999, Carlsson *et al.*, Int J Obes. 2012). It will be up to groups or consortia, specializing in the genetics of other autoimmune disorders, to examine in future studies whether our findings are significant also for these other disorders.

REVIEWERS' COMMENTS:

Reviewer #1 (Remarks to the Author):

All concerns have been sufficiently addressed.

Reviewer #2 (Remarks to the Author):

I think the Replies and the revised manuscript are acceptable.

Point-by-point reply to reviewers' comments

REVIEWERS' COMMENTS:

Reviewer #1 (Remarks to the Author):

All concerns have been sufficiently addressed.

We thank Reviewer#1 for positive comment of the revised manuscript.

Reviewer #2 (Remarks to the Author):

I think the Replies and the revised manuscript are acceptable.

We thank Reviewer#2 for positive comment of the revised manuscript.